# DeepResearch Bench: A Comprehensive Benchmark for Deep Research Agents

**Mingxuan Du**[1]*, **Benfeng Xu**[1,2]†, **Chiwei Zhu**[1], **Licheng Zhang**[1], **Xiaorui Wang**[2], **Zhendong Mao**[1]‡

[1]University of Science and Technology of China, [2]MetastoneTechnology, Beijing, China
`{dumingxuan, benfeng}@mail.ustc.edu.cn`

## Abstract

Deep Research Agents (DRAs) are emerging as one of the most practical classes of LLM-based agents. Given an open-ended research task, they find, analyze, and synthesize large numbers of online sources to produce a comprehensive report at the level of a research analyst. This can compress hours of manual desk research into minutes. However, a comprehensive benchmark for systematically evaluating the capabilities of these agents remains absent. To bridge this gap, we introduce **DeepResearch Bench**, a benchmark consisting of 100 PhD-level research tasks, each meticulously crafted by domain experts across 22 distinct fields. To evaluate DRAs comprehensively, we propose two complementary and fully automated methodologies. The first is a reference-based method with adaptive criteria to assess the quality of generated research reports. The second evaluates a DRA's information-retrieval and collection capabilities by assessing its effective citation count and overall citation accuracy. By conducting extensive human consistency experiments, we demonstrate that our evaluation methods are highly aligned with expert judges and faithfully reflect human judgments of quality differences among DRA-generated content. We have open-sourced DeepResearch Bench and key components of these frameworks at https://github.com/Ayanami0730/deep_-research_bench to accelerate the development of practical LLM-based agents.

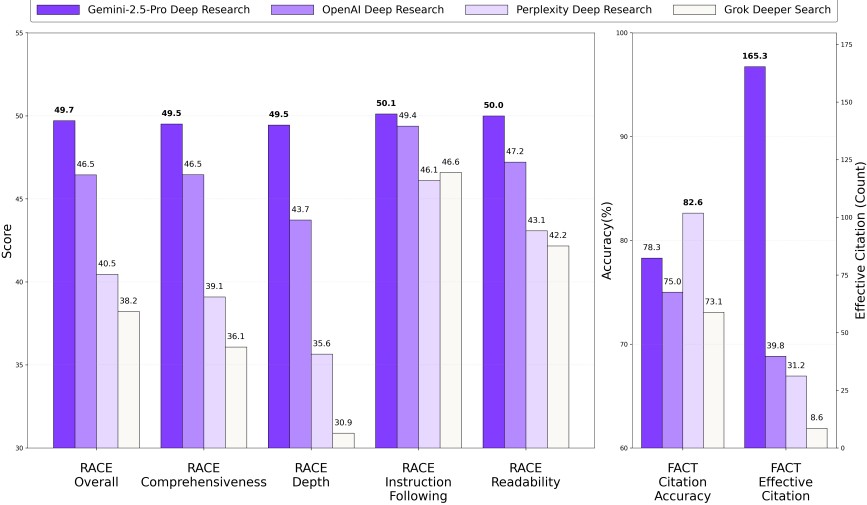

Figure 1: Overview of four Deep Research Agents' performance on DeepResearch Bench. The left figure shows the report quality scores achieved by the DRAs. The right figure shows each agent's citation accuracy and the average number of effective citations.

---

*Work done during the internship at Metastone.

†Project lead.

‡Corresponding author.

# 1 INTRODUCTION

We are now entering a new phase of AI (Yao, 2025), a period marked by comprehensive advances in Large Language Model (LLM) capabilities (DeepSeek-AI et al., 2025; OpenAI, 2024). These advancements enable the construction of LLM-based Agent systems designed to tackle increasingly complex tasks (Masterman et al., 2024; Hong et al., 2024; Yang et al., 2024). In this evolving landscape, defining tasks that genuinely reflect real-world demands and designing robust evaluation methodologies to measure the progress of these Agent systems are becoming critically important. Deep research represents one such well-defined task domain, with Deep Research Agents (DRAs) (Li et al., 2025; Zheng et al., 2025a; Schmidgall et al., 2025) emerging as the most widely utilized LLM-based agents today.

However, comprehensively evaluating DRAs is challenging. Because their internal reasoning and retrieval are opaque, the final report is the primary observable. Moreover, for complex research tasks, establishing a definitive ground truth is often infeasible.

These demanding evaluation requirements for DRAs pose a significant challenge for existing evaluation frameworks, which often fall short of offering a dedicated assessment of the multifaceted capabilities of such agents (Liu et al., 2023). Current benchmarks typically focus on assessing isolated capabilities—such as web browsing and information retrieval (Wei et al., 2025; Zhou et al., 2025; 2024), or generative abilities disconnected from real-time information acquisition (Que et al., 2024; Bai et al., 2024; Wu et al., 2025b).

To bridge this gap, we introduce **DeepResearch Bench**, a 100-task benchmark across 22 domains, with each task crafted and iteratively refined by domain experts. To reflect real research needs, we allocate per-domain task counts via a statistical analysis of over 96,000 user queries, following the pipeline in Figure 2(a).

Building on this dataset, we introduce two novel, highly human-aligned evaluation methodologies. The first one is a Reference-based and Adaptive Criteria-driven Evaluation framework with Dynamic Weighting (denoted as **RACE** for ease of subsequent reference), which targets the assessment of report generation quality. And the other one is a framework for Factual Abundance and Citation Trustworthiness (denoted as **FACT**), which focuses on evaluating information retrieval and citation accuracy. Overview results are shown in figure 1. **Furthermore, we believe these methodologies are not confined to deep research scenarios**; see Appendix J for broader discussion.

Our primary contributions are as follows:

- We present **DeepResearch Bench**, the first specialized benchmark for evaluating Deep Research Agents, built via large-scale analysis of real user queries and close collaboration with domain experts, balancing challenge while faithfully reflecting authentic user needs.
- We further propose **RACE** and **FACT**, two novel evaluation frameworks that respectively assess the report generation quality and the information retrieval abilities of Deep Research Agents.
- We conduct comprehensive human studies to validate the reliability of our frameworks, and will publicly release the benchmark and evaluation protocols upon acceptance to foster future research.

# 2 DEEPRESEARCH BENCH CONSTRUCTION

## 2.1 TOPIC DISTRIBUTION ANALYSIS

Deep Research Agents (DRAs) are intended to serve actual human research needs. Therefore, to effectively test their capabilities, the design of DeepResearch Bench is grounded in the real-world distribution of human research task demands. To obtain this distribution, we collected an in-house dataset of 96,147 raw user queries from interactions with web search-enabled Chatbots. To ensure user privacy, all raw query logs underwent rigorous anonymization. Further details of the in-house data are provided in Appendix B.

Following the pipeline shown in Figure 2(a), we then defined the concept of Deep Research tasks as problems requiring agents to conduct multiple rounds of web searches, gather information, perform

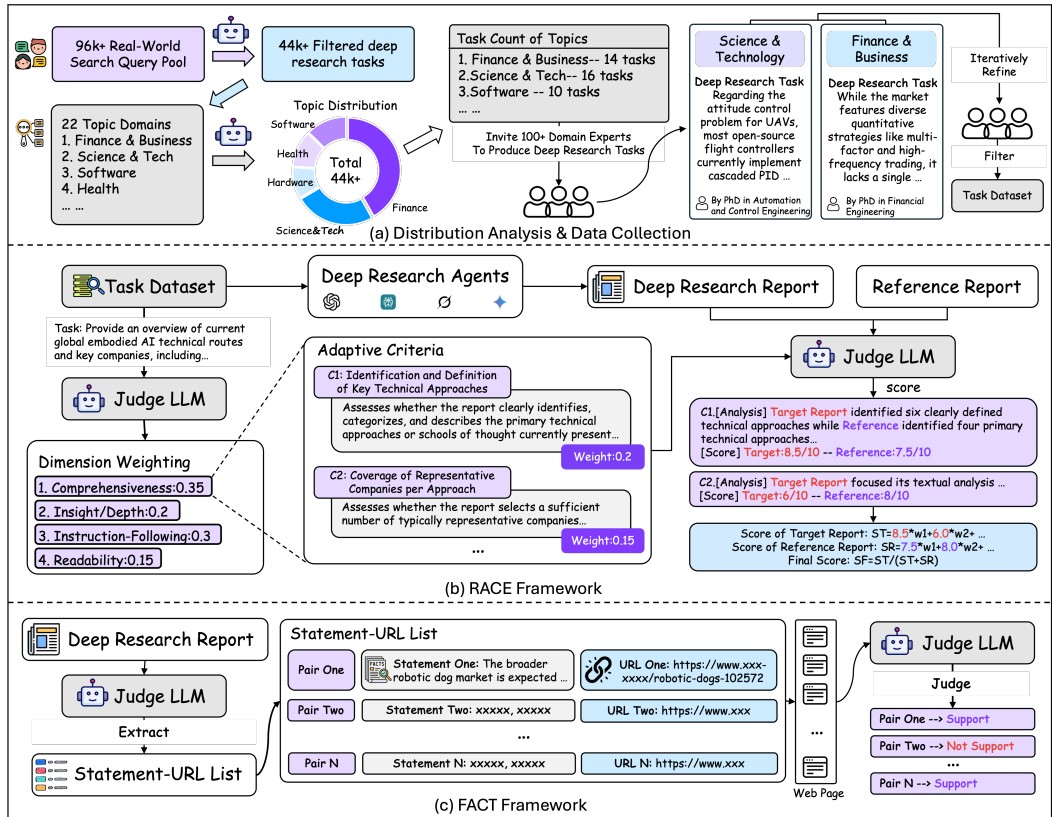

Figure 2: Overview of DeepResearch Bench. (a) Distribution analysis and dataset construction pipeline. (b) RACE framework. (c) FACT framework.

analysis, and produce high-quality reports. Guided by this definition, we employed DeepSeek-V3-0324 (DeepSeek-AI, 2025) to conduct filtering, identifying queries that aligned with our deep research requirements. This process ultimately yielded a dataset of 44,019 queries conforming to our deep research task definition.

To categorize the deep research queries, we adopted the topic taxonomy proposed by WebOrganizer (Wettig et al., 2025), selecting 22 distinct topic domains for this classification. We then employed DeepSeek-V3-0324 to classify these 44,019 queries into the 22 selected topic domains. By statistically aggregating the LLM's classification results, we obtained the distribution of these queries across the various topics. This distribution, visualized in Figure 3, indicates the real-world user demand for deep research within these domains.

## 2.2 BENCHMARK TASK COLLECTION

Guided by the observed user-demand distribution, we set the target number of tasks per domain and proportionally compressed it to a final set of 100 tasks (50 Chinese, 50 English), preserving the topical balance. The bilingual design reflects the multilingual nature of real-world deep research demands. The dataset size was limited because running a single deep-research task is resource-intensive; moreover, many frontier benchmarks, such as xbench-DeepSearch (Chen et al., 2025a) and Mind2Web 2 (Gou et al., 2025), contain roughly one hundred tasks, indicating that this scale represents a practical trade-off between quality and stability.

Once the target task count for each topic domain was determined, our focus shifted to constructing research tasks that are both highly challenging and firmly grounded in authentic research demands. We invited PhD holders and senior practitioners with over five years of relevant domain experience to propose candidate tasks. It is important to clarify that these 100 benchmark tasks were independently created by domain experts based on the target allocations, rather than being selected from the

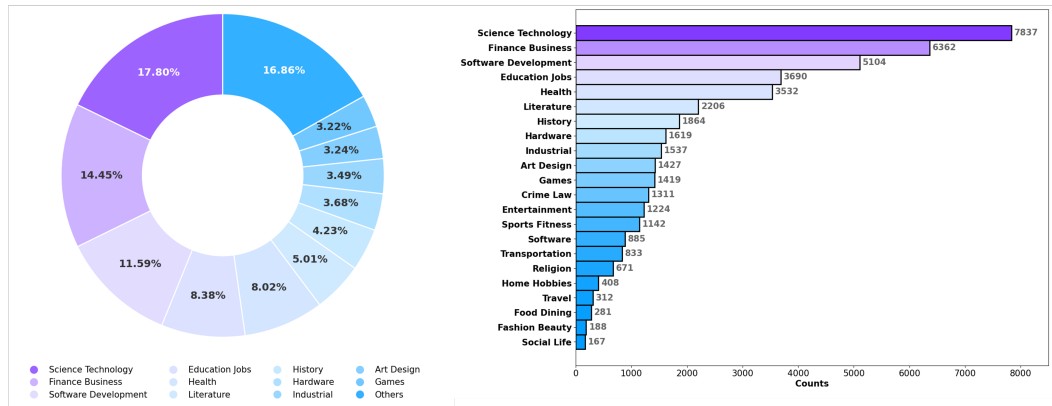

Figure 3: Topic distribution of the filtered deep-research tasks. Left: donut chart showing topic share. Right: bar chart of absolute task counts in 22 domains.

44,019 filtered queries. The 44K samples served solely to inform the topic distribution, ensuring our benchmark reflects real-world demand patterns. Examples are shown in Figure 4. All submissions underwent manual screening by our team to verify their quality, clarity, complexity, and alignment with our definition of deep research. This rigorous vetting process resulted in the 100 high-quality benchmark tasks that constitute DeepResearch Bench.

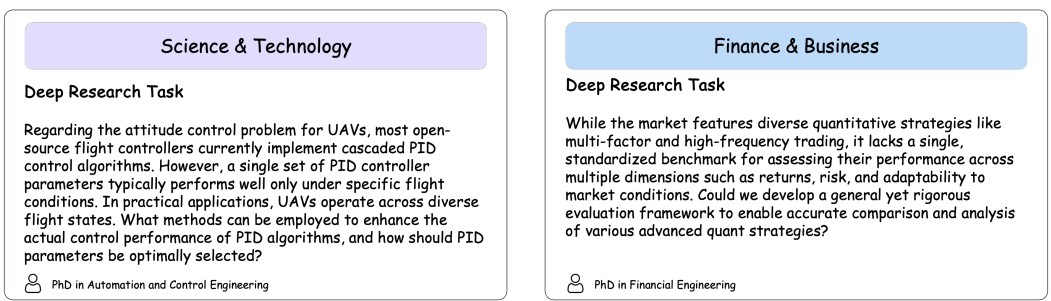

Figure 4: Two example tasks from DeepResearch Bench.

## 3 EVALUATION METHODOLOGY

Our evaluation methodology focuses on two critical aspects: their capabilities in information retrieval and collection, and the quality of the final reports they generate. To assess these respective dimensions, we developed two complementary frameworks, RACE and FACT.

### 3.1 RACE: A FRAMEWORK FOR REPORT QUALITY EVALUATION

Evaluating long-form research reports presents significant challenges. Existing approaches using fixed checklists Que et al. (2024) or static rubrics Shao et al. (2024); Bai et al. (2024) struggle to adapt to diverse tasks, specialized domains, and nuanced quality aspects of deep research tasks. To address this, we introduce our Reference-based Adaptive Criteria-driven Evaluation framework with Dynamic Weighting (**RACE**), leveraging the LLM-as-a-Judge method Zheng et al. (2023). RACE offers a more adaptive and robust evaluation by first dynamically generating task-specific weights and criteria. It then employs a reference-based scoring approach, comparing the target report to a high-quality reference. Finally, a relative score is computed to assess the target report's quality.

**Dynamic Weight & Adaptive Criteria Generation.** Directly prompting LLMs to generate task-specific criteria from scratch can lead to results that deviate significantly from the intended assessment goals. Following Google's Gemini Deep Research guidance (Google, 2025), we adopt similar

high-level design principles and establish four orthogonal top-level dimensions based on domain expertise: **Comprehensiveness (COMP)**, **Insight/Depth (DEPTH)**, **Instruction-Following (INST)**, and **Readability (READ)**. Detailed definitions are provided in Appendix C.

As illustrated in Figure 2(b), for each task $t$, the Judge LLM is prompted to produce task-specific weights $W_d$ for the four dimensions $d \in \{\text{COMP}, \text{DEPTH}, \text{INST}, \text{READ}\}$. These weights ensure the evaluation aligns with the task's intent. Subsequently, for each dimension $d$, the Judge LLM generates a set of tailored criteria $\{c_{d,k}\}$ with corresponding weights $\{w_{d,k}\}$ (where $\sum_{k=1}^{K_d} w_{d,k} = 1$), which are clear and actionable for evaluating the report within that dimension. Importantly, once generated for a task, these criteria and weights remain fixed across all DRA evaluations for that task, ensuring consistent and fair comparison. This query-based approach also enables future work to easily scale up the testset by generating criteria for new tasks.

**Reference-Based Scoring.** Preliminary experiments indicated that scoring reports in isolation often yields insufficiently discriminative results; models tend to assign uniformly high scores, masking genuine quality variations. To mitigate this, RACE adopts a reference-based scoring strategy. For each task $t$, a high-quality report $R_{\text{ref}}$ is selected as a reference. All generated criteria $\{c_{d,k}\}$ across all dimensions are aggregated into a comprehensive list $\mathcal{C}_t$. The Judge LLM then analyzes the target report $R_{\text{tgt}}$ and the reference report $R_{\text{ref}}$ against each criterion $c \in \mathcal{C}_t$. This yields lists of scores for both reports for each criterion, which are then used for final score calculation:

$$(\{s_{\text{tgt},c}\}_{c \in \mathcal{C}_t}, \{s_{\text{ref},c}\}_{c \in \mathcal{C}_t}) = \text{JudgeLLM}(t, R_{\text{tgt}}, R_{\text{ref}}, \mathcal{C}_t) \tag{1}$$

**Overall Score Calculation.** Finally, we compute the overall quality score of the target report. First, dimension-level scores $S_d(R)$ are calculated by weighting criterion-level scores $s_{R,c_{d,k}}$ with criterion weights $w_{d,k}$. Second, these $S_d(R)$ scores are combined using the task-specific dimension weights $W_d$ to produce intermediate overall scores $S_{\text{int}}(R)$. Finally, the target report's score $S_{\text{final}}(R_{\text{tgt}})$ is determined relative to the reference report's score:

$$S_{\text{final}}(R_{\text{tgt}}) = \frac{S_{\text{int}}(R_{\text{tgt}})}{S_{\text{int}}(R_{\text{tgt}}) + S_{\text{int}}(R_{\text{ref}})} \tag{2}$$

### 3.2 FACT: A FRAMEWORK FOR WEB RETRIEVAL EVALUATION

To assess the factual grounding of report content and the agent's effectiveness in retrieving and utilizing web-based information, we introduce a framework for Factual Abundance and Citation Trustworthiness(**FACT**). As illustrated in Figure 2(c), let $T$ denote the set of all tasks in the benchmark. For each task $t \in T$, FACT evaluates DRAs through the following automated steps:

**Statement-URL Pair Extraction and Deduplication.** We employ a Judge LLM to extract discrete statements from the DRA-generated report for task $t$, along with their corresponding cited source URLs. Let $P_t$ denote the initial set of all extracted statement-URL pairs for task $t$. The Judge LLM then examines $P_t$ to identify cases where multiple statements associated with the same URL describe exactly the same fact. In such cases, only one representative pair is retained. We denote the deduplicated set as $U_t$, with $N_{u,t} = |U_t|$ representing the total number of unique statement-URL pairs for task $t$.

**Support Judgment.** Each unique statement-URL pair in $U_t$ undergoes a support evaluation. We retrieve the textual content of the cited webpage using the Jina Reader API, then the Judge LLM assesses whether this content provides sufficient evidence for the statement. This yields a binary judgment: 'support' or 'not support'. Let $N_{s,t}$ denote the number of pairs in $U_t$ that are judged as 'support' for task $t$.

**Calculation of Citation Metrics.** Based on these support judgments across all tasks, we calculate two primary metrics:

**Citation Accuracy (C. Acc.)** measures the precision of an agent's citations, reflecting its ability to correctly ground statements with appropriate sources. For each task $t$, we first compute the per-task accuracy as $Acc_t = N_{s,t}/N_{u,t}$ (with $Acc_t = 0$ if $N_{u,t} = 0$). The overall Citation Accuracy is then

the average across all tasks:

$$\text{C. Acc.} = \frac{1}{|T|} \sum_{t \in T} Acc_t \tag{3}$$

**Average Effective Citations per Task (E. Cit.)** quantifies the average number of verifiably supported statements an agent generates per task. It sums the supported pairs across all tasks and divides by the total number of tasks:

$$\text{E. Cit.} = \frac{\sum_{t \in T} N_{s,t}}{|T|} \tag{4}$$

## 4 EXPERIMENTS

### 4.1 EXPERIMENTAL SETUP

**Implementation Details** When employing the RACE framework, a pre-processing step involves cleaning citation formatting from the generated reports, as overly lengthy or complex citation styles can adversely affect the Judge LLM's scoring process. For RACE evaluation tasks, we utilize Gemini-2.5-pro as the Judge LLM. As for the FACT framework, Gemini-2.5-flash is employed for both Statement-URL pair extraction and support judgment, which is sufficient in capabilities while more economic for the token-consuming citation verification task. The reference reports used in RACE's scoring methodology were selected from deep research articles generated by the Gemini-2.5-pro-based Deep Research, as available in April 2025. All main results reported in Table 1 are evaluated on the complete set of 100 tasks from DeepResearch Bench.

**Evaluated Models** In our work, we broadly evaluate leading commercial Deep Research Agents, including OpenAI Deep Research, Gemini-2.5-pro-based Deep Research and so on. Due to the lack of transparency regarding the iteration cycles of these commercial products, we specify the data collection timeframes in Appendix E. In the open-source domain, we evaluate two representative RL-based systems with open-source frameworks: (1) DeepResearcher Zheng et al. (2025b), trained via reinforcement learning in real-world web environments, and (2) Tongyi DeepResearch Fu et al. (2025), leveraging RL training with Qwen3 as the base model. Both are locally deployed following their official open-source implementations. Additionally, we evaluate LangChain's Open Deep Research (ODR) (LangChain et al., 2025), an open-source framework with frontier LLM as backbone. We test it with both GPT-4.1 and GPT-5 as the backbone LLMs, with configurations detailed in Appendix F. We also evaluate strong LLMs with built-in search tools under standardized conditions by setting the search_context_size to high; see Appendix G.2.

### 4.2 MAIN RESULTS

#### 4.2.1 EVALUATION ON RACE FRAMEWORK

As shown in Table 1, within the DRA category, Gemini-2.5-Pro Deep Research achieved the highest overall performance, while OpenAI Deep Research also demonstrated strong capabilities. Notably, the open-source LangChain Open Deep Research (ODR) further exhibited competitive results, surpassing several proprietary DRAs in our evaluation. When upgraded from GPT-4.1 to GPT-5 as the backbone, LangChain ODR achieves an overall score of 50.60, surpassing even Gemini-2.5-Pro Deep Research, demonstrating that open-source frameworks can achieve competitive results with state-of-the-art proprietary models when powered by advanced LLMs.

The scores for these top agents are relatively close, which is characteristic of the reference-based relative scoring employed by RACE. However, this should not be concerning, as subsequent experiments 4.3 revealed a strong linear correlation between these RACE scores and human judgments, suggesting the framework effectively captures meaningful performance differences between models. In fact, these scores are highly linearly correlated with human evaluations, just mapped to a different reference frame (similar to how scores of 45 and 50 versus human scores of 90 and 100). Therefore, we should focus on rankings and proportional differences between scores rather than absolute score values.

For fully open-source DR systems, we observe significant performance gaps. Both DeepResearcher Zheng et al. (2025b) and Tongyi DeepResearch Fu et al. (2025) are trained via RL with

Table 1: Overall evaluation results of DeepResearch Bench. **Bold** denotes the highest score in each column for Deep Research Agents (and for LLM with Search Tools within their respective section). Underlined denotes the second highest.

| Model | RACE | | | | | FACT | |
|---|---|---|---|---|---|---|---|
| | Overall | Comp. | Depth | Inst. | Read. | C. Acc. | E. Cit. |
| *Open-Source LLM + Open-Source Agent Framework* | | | | | | | |
| DeepResearcher | 10.77 | 8.66 | 5.96 | 15.57 | 15.65 | – | – |
| Tongyi DeepResearch | 40.46 | 39.46 | 34.44 | 46.22 | 44.27 | – | – |
| *Proprietary LLM + Open-Source Agent Framework* | | | | | | | |
| LangChain Open DeepResearch (GPT-5) | 50.60 | 50.06 | 50.76 | 51.31 | 49.72 | 32.94 | 21.06 |
| LangChain Open DeepResearch (GPT-4.1) | 43.44 | 42.97 | 39.17 | 48.09 | 45.22 | 49.10 | 29.49 |
| *Proprietary Deep Research Agent* | | | | | | | |
| Claude Research | 45.00 | 45.34 | 42.79 | 47.58 | 44.66 | – | – |
| Grok Deeper Search | 38.22 | 36.08 | 30.89 | 46.59 | 42.17 | 73.08 | 8.58 |
| Perplexity Deep Research | 40.46 | 39.10 | 35.65 | 46.11 | 43.08 | **82.63** | 31.20 |
| Doubao Deep Research | 44.34 | 44.84 | 40.56 | 47.95 | 44.69 | 52.86 | 52.62 |
| Gemini-2.5-Pro Deep Research | **49.71** | **49.51** | **49.45** | **50.12** | **50.00** | 78.30 | **165.34** |
| Kimi Researcher | 44.64 | 44.96 | 41.97 | 47.14 | 45.59 | – | – |
| OpenAI Deep Research | 46.45 | 46.46 | 43.73 | 49.39 | 47.22 | 75.01 | 39.79 |
| *Proprietary LLM + Built-in Search Tools* | | | | | | | |
| Claude-3.7-Sonnet w/Search | **40.67** | **38.99** | **37.66** | **45.77** | 41.46 | 93.68 | 32.48 |
| Claude-3.5-Sonnet w/Search | 28.48 | 24.82 | 22.82 | 35.12 | 35.08 | **94.04** | 9.78 |
| Perplexity-Sonar-Reasoning-Pro | 40.22 | 37.38 | 36.11 | 45.66 | **44.74** | 39.36 | 8.35 |
| Perplexity-Sonar-Reasoning | 40.18 | 37.14 | 36.73 | 45.15 | 44.35 | 48.67 | 11.34 |
| Perplexity-Sonar-Pro | 38.93 | 36.38 | 34.26 | 44.70 | 43.35 | 78.66 | 14.74 |
| Perplexity-Sonar | 34.54 | 30.95 | 27.51 | 42.33 | 41.60 | 74.42 | 8.67 |
| Gemini-2.5-Pro-Grounding | 35.12 | 34.06 | 29.79 | 41.67 | 37.16 | 81.81 | **32.88** |
| Gemini-2.5-Flash-Grounding | 32.39 | 31.63 | 26.73 | 38.82 | 34.48 | 81.92 | 31.08 |
| GPT-4o-Search-Preview | 35.10 | 31.99 | 27.57 | 43.17 | 41.23 | 88.41 | 4.79 |
| GPT-4o-Mini-Search-Preview | 31.55 | 27.38 | 22.64 | 40.67 | 39.91 | 84.98 | 4.95 |
| GPT-4.1 w/Search | 33.46 | 29.42 | 25.38 | 42.33 | 40.77 | 87.83 | 4.42 |
| GPT-4.1-mini w/Search | 30.26 | 26.05 | 20.75 | 39.65 | 39.33 | 84.58 | 4.35 |

open-source frameworks that can be locally deployed. DeepResearcher achieves an overall score of 10.77, significantly lower than other systems. Through manual inspection, we found that it frequently fails to generate complete, well-structured reports, which substantially impacts its RACE scores. In contrast, Tongyi DeepResearch achieves an overall score of 40.46, comparable to Perplexity Deep Research among proprietary DRAs. This demonstrates that with appropriate RL training, open-source systems can approach the performance of commercial products, though challenges remain in generating high-quality long-form reports consistently.

Moreover, traditional LLMs with built-in search (often limited to single-round or a few simple search turns) now struggle to compete with modern DRAs under identical evaluation settings.

### 4.2.2 EVALUATION ON FACT FRAMEWORK

Viewing evaluation results by FACT in Table 1, Deep Research Agents (except Grok) tend to include more Effective Citations than LLMs with Search Tools. Notably, Gemini-2.5-Pro Deep Research achieved an average of 165.34 effective citations in its final reports, significantly outperforming other models. This suggests that it can retrieve and integrate more information from a larger amount of evidence, potentially enabled by a longer context window and stronger context understanding. However, its citation accuracy is lower than that of Perplexity Deep Research and far below Claude-

3.7 w/Search, indicating a trade-off between citation accuracy and effective citation counts to some extent.

We also note that Claude Research and Kimi Research do not have FACT scores because we were unable to parse citation links from their official UI. For open-source RL-based systems (DeepResearcher and Tongyi DeepResearch), FACT scores are unavailable because these systems failed to follow instructions to generate citations in the required format, making citation extraction and verification infeasible. Meanwhile, LangChain Open Deep Research and Doubao show relatively low FACT scores. For LangChain Open Deep Research, the low citation accuracy can be attributed to its multi-agent architecture with a dedicated summarization sub-agent, which causes misalignment between citations and the retrieved content during the summarization process. For Doubao, based on our inspection, many webpages reachable by its built-in browse tools were inaccessible to our Jina-based crawling pipeline, and even for accessible pages, the fetched contents sometimes differed, which likely affected the measured metrics.

Overall, the FACT framework is designed to offer a complementary observation to RACE; given the lack of transparency and variability of built-in search/browse tools across DRAs, we use FACT as an observational dimension for analysis, while our benchmark's overall rankings rely on RACE scores.

### 4.3 HUMAN CONSISTENCY

Evaluating the quality of deep research reports remains an open-ended task. Therefore, to validate the effectiveness of our proposed RACE framework, we must rely on assessing its human consistency. We conducted experiments using 50 Chinese tasks from DeepResearch Bench, with reports generated by four distinct agents. For each task, three domain-expert annotators scored these reports. Further details are provided in Appendix G.1.

#### 4.3.1 HUMAN DATA COLLECTION

To gather human judgments, we recruited 70+ annotators with Master's degrees and relevant domain expertise. Using a custom interface, they evaluated reports across four dimensions and overall performance, guided only by basic scoring criteria to minimize bias. Each annotator was limited to three queries maximum to ensure diverse perspectives.

#### 4.3.2 EVALUATION METRICS

To validate the consistency between evaluation methods and human judgment, we designed four metrics that quantify different aspects of alignment with human evaluations. The detailed calculation processes for all metrics are provided in Appendix H.

**Pairwise Agreement Rate (PAR)** This metric measures how often our evaluation method's preferences match human experts' preferences when comparing pairs of reports.

**Overall Pearson Correlation (OPC)** This metric quantifies the linear relationship between average model scores from our evaluation method and those from human experts.

**Filtered Average Pearson & Spearman Correlation** To mitigate per-task noise, we filter out tasks with low inter-rater agreement (ICC < 0) and then compute two metrics on the remaining subset: the Filtered Average Pearson Correlation (FAP) and the Filtered Average Spearman Correlation (FAS). Detailed definitions and formulas are provided in Appendix H.

#### 4.3.3 COMPARISON OF DIFFERENT EVALUATION METHODS

Given that existing evaluation methods are generally unsuitable for assessing DRAs, we compare RACE(Full) and several ablation variants against a Vanilla Prompt baseline (direct scoring by the Judge LLM). As shown in Table 3, RACE(Full) achieves the best overall performance, significantly exceeding the baseline and other variants. Notably, its Pairwise Agreement Rate also surpasses human inter-agreement, indicating reliable and efficient human-aligned evaluation. We further include robustness experiments on reference selection, article length, and judge model choice in Appendix I.

Table 2: Comparison of human consistency scores across different evaluation methods. Prefixed with '-', indicating removal of specific components from the full framework. Best scores for each metric among automated methods are in **bold**.

| Evaluation Method | PAR | OPC | FAP | FAS | Overall Score |
|---|---|---|---|---|---|
| Vanilla Prompt | 58.89 | 98.89 | 40.30 | 43.75 | 60.46 |
| **RACE(Full)** | **71.33** | 99.54 | **60.24** | **59.12** | **72.56** |
|   - No Criteria Weights | 70.67 | 99.62 | 59.83 | 56.27 | 71.60 |
|   - No Dim Weights | 70.89 | 99.54 | 60.11 | 57.22 | 71.94 |
|   - No Weights | 71.11 | **99.69** | 59.46 | 58.17 | 72.11 |
|   - No Reference | 66.56 | 97.46 | 57.51 | 51.23 | 68.19 |
| Reverse Position | 69.56 | 97.20 | 56.75 | 55.49 | 69.75 |
| Static Criteria | 68.33 | 98.73 | 57.86 | 57.70 | 70.65 |
| Human Inter-Agreement | 68.44 | - | - | - | - |

Table 3: Comparison of human consistency scores, and average cost per task using different Judge LLMs within the RACE(Full) framework. The best for each metric are in **bold**

| Judge LLM | License | PAR | OPC | FAP | FAS | Overall | Cost ($) |
|---|---|---|---|---|---|---|---|
| Gemini 2.5 Pro | Proprietary | **71.33** | **99.54** | **60.24** | 59.12 | **72.56** | 0.13 |
| o3 | Proprietary | 68.11 | 96.22 | 57.64 | 52.36 | 68.58 | 0.37 |
| o4-mini | Proprietary | 70.89 | 97.06 | 59.54 | 59.02 | 71.63 | **0.04** |
| Claude 3.7 Sonnet | Proprietary | 70.78 | 96.53 | 58.22 | **63.61** | 72.28 | 0.47 |
| Qwen3-235B-Thinking | Apache 2.0 | 70.78 | 84.47 | 56.80 | 56.94 | 67.25 | – |

#### 4.3.4 COMPARISON OF DIFFERENT JUDGE LLM

Leveraging the RACE framework, we further compare the performance and cost of several leading LLMs when used as the Judge LLM. As detailed in Table 3, Gemini 2.5 Pro achieves the best overall performance and maintains a competitive average cost ($0.13 per query), only higher than that of o4-mini. In addition, we experimented with an open-source alternative, Qwen3-235B-A22B-Thinking-2507, as the Judge LLM. While its human consistency lags behind closed-source models, the gap is not large, suggesting it can serve as a feasible open-source substitute. Moreover, with Qwen3 as the Judge, the relative ranking across the four DRAs remains consistent with the Gemini-based results, indicating that RACE is robust to the choice of Judge backbone. Further details on using open-source judges are provided in Appendix I.3. To balance performance and cost in our main results, we selected Gemini 2.5 Pro as the Judge LLM in our final framework.

## 5 RELATED WORK

**LLM-based Agent Evaluation** With the comprehensive advancement of LLM capabilities, LLM-based Agents are increasingly being applied to real-world scenarios Mon-Williams et al. (2025); Wang et al. (2025), promising to alter many aspects of daily life and professional work significantly. Yao's blog Yao (2025) highlights that defining more realistic problems and designing novel evaluation methods are critical for constructing more practical AI Agent systems. Numerous evaluations have already been designed specifically for Agents, targeting diverse capabilities. These include evaluations for agents in scientific domains Chan et al. (2025); Laurent et al. (2024); Mitchener et al. (2025); Chen et al. (2025b), creative writing Wu et al. (2025b); Bai et al. (2024); Que et al. (2024), code generation and software engineering Jimenez et al. (2024); Zhuo et al. (2025); Quan et al. (2025); Jain et al. (2024); Xiao et al. (2025), and in their roles as human assistants, often enhanced by capabilities such as web browsing and tool-use Wei et al. (2025); Zhou et al. (2025); Yan et al. (2024); Deng et al. (2024); Wang et al. (2024). Closest to our setting, Xu et al. (2025) focuses on a single scientific domain and does not reflect real-world user demand, while Bosse et al. (2025) adopts an offline RetroSearch setting and reports process-oriented metrics (e.g., hallucination/tool usage) rather than evaluating report quality or citation fidelity. This perspective underscores our belief that constructing benchmarks specifically designed for Deep Research Agent, grounded in

real-world scenarios, alongside developing human-aligned evaluation methods, is urgently needed to guide the development of AI agent systems.

**Deep Research Agent** After the release of Deep Research Agents (DRAs) by OpenAI OpenAI (2025) and Google's Gemini Google (2025), such agents have attracted significant attention and have become one of the most widely deployed LLM-based agent categories. Subsequently, related works LangChain et al. (2025); Li et al. (2025); Zheng et al. (2025a) quickly followed up, also introducing their own designed DRA frameworks. However, the field still lacks a standardized evaluation methodology for these DRAs, preventing meaningful comparative analysis of their capabilities. Among these works, some use QA datasets Phan et al. (2025); Mialon et al. (2023); Wu et al. (2025a) as evaluation metrics, but this approach neither aligns with real-world DRA applications nor comprehensively assesses their broader capabilities. Others employ the LLM-as-a-judge methodology Zheng et al. (2023), yet these efforts lack both a comprehensive framework design and verification of human consistency. In contrast, our DeepResearch Bench addresses this gap by providing a systematic, unified evaluation method with strong human consistency, supporting subsequent DRA development and assessment.

## 6 CONCLUSION

In this work, we introduce DeepResearch Bench, the first comprehensive benchmark for evaluating the report generation and web retrieval capabilities of Deep Research Agents. Comprising 100 high-quality research tasks across 22 distinct domains, this benchmark is meticulously curated to reflect authentic user needs. Our key evaluation frameworks, RACE and FACT, have demonstrated high consistency with human judgments, affirming their reliability. We hope DeepResearch Bench will guide developers and researchers in constructing more powerful and human-centric AI agent systems, truly addressing genuine user requirements.

## ETHICS STATEMENT

All authors affirm adherence to the ICLR Code of Ethics. This work does not involve personally identifiable information or sensitive attributes. Our in-house query logs were rigorously anonymized prior to processing (removing user identifiers, IPs, and session metadata). Human studies were conducted with experienced annotators, who were compensated fairly; no protected or vulnerable populations were involved. We report results transparently and avoid claims beyond empirical evidence. Potential risks include misuse of agent outputs and propagation of web inaccuracies; our FACT framework and standardized citation formatting are designed to encourage faithful grounding and make verification easier. We disclose no conflicts of interest or external sponsorship that would bias the findings.

## REPRODUCIBILITY STATEMENT

We provide all essential details to reproduce our results. The benchmark construction pipeline (topic taxonomy, filtering and classification steps), evaluation prompts, and hyperparameters are documented in the main text and Appendix (e.g., Sections 2, 3, and implementation details in Section 4.1). Exact Judge LLM configurations for RACE/FACT, including dynamic weighting and criteria generation prompts, are included in the Appendix. All scripts and data used in this paper are included in the supplementary materials, which also contain screenshots of the annotation interface and the annotator instructions/guidelines to facilitate faithful reproduction.

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

## A   LIMITATIONS

While **DeepResearch Bench** and the RACE/FACT frameworks offer a comprehensive evaluation of Deep Research Agents, several limitations remain. (1) **Benchmark scale**: Curating expert-level, distribution-grounded tasks is labor-intensive; our 100-task set prioritizes quality and topic balance over scale. (2) **Domain coverage**: Despite multi-domain design, residual coverage bias may exist; future versions will broaden reviewer pools and domains. (3) **Human evaluation throughput**: Expert judging is costly, limiting sample sizes; we plan larger studies to further tighten confidence intervals. (4) **Tooling opacity**: FACT depends on each system's built-in search/browse stack and external fetchers; differences in reachability and page variants can affect effective-citation measurement. (5) **Judge dependence**: RACE relies on a Judge LLM via proprietary APIs. In the paper and Appendix, we report results with open-source judges (e.g., Qwen3) and find they can serve as practical substitutes to a certain extent, offering a more open alternative.

## B   IN-HOUSE DATA DETAILS

At the time of constructing DeepResearch Bench, there was no open-source dataset that directly captured real user interactions with production DRAs. Consequently, to support the analysis of realistic topic distributions, we resorted to an in-house log of user interactions with a search-augmented chatbot. We then applied a multi-stage post-processing pipeline, including anonymization, filtering to extract deep-research style tasks, and topic categorization, to approximate the domain distribution of real-world DRA tasks.

The collected queries cover a wide range of real information needs, for example: "analyze the time-series trend of used-car prices for a specific brand in my city," and "investigate a product's R&D team, development timeline, and estimate its ARR." This diversity, together with careful identification of deep-research tasks, leads us to believe that the resulting distribution is a close proxy to real user demand.

As the closest public reference, we also examined the recently released *Search Arena 24K* dataset. Using exactly the same filtering and categorization pipeline as in the main paper, we derived a deep-research domain distribution from Search Arena 24K and compared it against our in-house distribution (Table 4).

As we can see, the two distributions share the same top-5 domains and are overall similar across major categories, which supports the reasonableness of using our in-house data as a surrogate for real-world DRA usage.

Table 4: Major topic distribution comparison (%) between our in-house data and the open-source Search Arena 24K dataset.

| Category | Sci & Tech | Fin & Biz | Soft Dev | Edu & Jobs | Health | Literature |
|---|---|---|---|---|---|---|
| Our In-house Data | 17.80 | 14.45 | 11.59 | 8.38 | 8.02 | 5.01 |
| Search Arena 24K | 13.68 | 17.74 | 15.81 | 4.97 | 10.47 | 4.54 |

| Category | History | Hardware | Industrial | Art & Design | Games |
|---|---|---|---|---|---|
| Our In-house Data | 4.23 | 3.68 | 3.49 | 3.24 | 3.22 |
| Search Arena 24K | 4.42 | 3.19 | 4.10 | 1.54 | 2.57 |

## C   DIMENSION DEFINITIONS

The RACE framework evaluates research reports based on four top-level dimensions. Their definitions are provided in Table 5.

Table 5: Definitions of Core Evaluation Dimensions for Report Quality

| Dimension | Description |
|---|---|
| **Comprehensiveness** (COMP) | Article covers key areas of the industry, ensures overall understanding, and does not omit important parts. |
| **Insight/Depth** (DEPTH) | Article deeply analyzes causes, impacts, and trends, providing valuable insights. |
| **Instruction-Following/Relevance** (INST) | Article closely follows the research topic and directly answers questions. |
| **Readability** (READ) | Article has a clear structure, fluent language, and is easy to understand. |

## D  JUDGE LLM SELECTION FOR THE FACT FRAMEWORK

The FACT framework employs a Judge LLM for crucial automated steps: Statement-URL Pair Extraction and Deduplication, followed by Support Judgment. The selection of this Judge LLM is pivotal, aiming to balance evaluation accuracy with operational costs, especially given the significant token consumption inherent in these processes. To determine an optimal model for these tasks, we specifically evaluated `Gemini-2.5-Flash`. Its judgments were compared against human evaluations on a randomly sampled set of 100 statement-URL pairs derived from our benchmark tasks. This comparison demonstrated strong agreement with human annotators: `Gemini-2.5-Flash`'s judgment aligned with human 'support' determinations in 96% of cases and with 'not support' determinations in 92% of cases.

We further find out that the accuracy of `Gemini-2.5-Flash` in these FACT-specific evaluation steps is very close to that of `Gemini-2.5-Pro`. The operations within the FACT framework (such as extracting statements from full reports and analyzing webpage content for support) are known to be token-intensive, making cost-effectiveness a critical consideration. Since `Gemini-2.5-Flash` demonstrated comparable accuracy to `Gemini-2.5-Pro` for these specific tasks but at a more advantageous cost, we select `Gemini-2.5-Flash` as the Judge LLM for the FACT framework. This choice enables us to maintain high evaluation reliability while managing operational costs effectively.

## E  DATA COLLECTION TIMEFRAMES

The data for the commercial models evaluated in this paper were collected during specific timeframes in 2025, as detailed in Table 6. These dates indicate when the model outputs used in our experiments were generated.

Table 6: Data Collection Timeframes for Evaluated Models (2025)

| Model Category / Provider Group | Data Collection Date Range |
|---|---|
| *Deep Research Agents (DRAs)* | |
| OpenAI Deep Research | April 1 – May 8 |
| Gemini 2.5 Pro Deep Research | April 27 – April 29 |
| Perplexity Deep Research | April 1 – April 29 |
| Grok Deeper Search | April 27 – April 29 |
| Claude Research | June 23 – June 25 |
| Doubao Deep Research | June 29 – July 1 |
| Kimi Researcher | June 29 – July 1 |
| LangChain Open Deep Research | June 29 – July 1 |
| *LLM with Search Tools (Grouped by Provider)* | |
| Claude Models (w/Search) | May 12 – May 13 |
| Perplexity Models | May 11 – May 12 |
| GPT Models (w/Search) | May 11 – May 12 |
| Gemini Models (Grounding/w/Search) | May 12 – May 13 |

## F   OPEN DEEP RESEARCH REPRODUCTION DETAILS

We evaluate the open-source LangChain Open Deep Research (ODR) implementation using its default settings. Specifically, the research model is set to GPT-4.1 and the summarization model is set to GPT-4.1-mini, following the project's recommended defaults. Unless otherwise stated, no additional modifications are applied to the configuration.

## G   EXPERIMENT DETAIL

### G.1   HUMAN EVALUATION EFFORT

The human evaluation process required considerable time investment to ensure thorough assessment. On average, each expert annotator spent approximately 1.5 hours per query to evaluate the reports from the four agents. This meticulous process resulted in a total of 225 person-hours of human evaluation across all tasks and annotators. This substantial effort provided a robust and reliable dataset of human judgments, forming the basis for our human consistency analysis.

### G.2   CONFIGURATION OF LLMS WITH WEB SEARCH TOOLS

To ensure a standardized and comparable evaluation environment for Large Language Models (LLMs) equipped with built-in web search tools, the following configurations were uniformly applied:

- **Thinking Budget/Computational Resources:** For models that support a configurable "thinking budget" or a similar computational resource limit for generation, this was uniformly set to a high value, equivalent to 16,000 tokens where applicable. This allowed models ample processing capacity for complex queries.

- **Search Context Size:** In cases where models offered a parameter to control the amount of information retrieved and utilized from web searches (e.g., the `search_context_-size` option as found in the Perplexity AI API), this was consistently set to "high". This configuration aimed to maximize the contextual information available to the LLM from its search activities.

- **Maximum Search Iterations:** The maximum number of web search queries, or "search turns," permitted during the generation process was standardized to five for all LLMs that provided such a configurable limit. This ensured a comparable depth of web exploration across these models.

- **Output Length:** To accommodate potentially comprehensive responses while maintaining consistency, the maximum output token limit for all LLMs was set to 36,000 tokens. If a model's inherent maximum output capacity was less than this 36,000-token threshold, its specific native maximum limit was adhered to.

- **Citation Formatting and Standardization:** A critical aspect of our methodology was the standardization of citation presentation to facilitate consistent downstream evaluation, particularly when employing frameworks like FACT for factual assessment. Citations as provided by each LLM were parsed in accordance with their respective official API documentation. Subsequently, the generated reports were systematically restructured: citation markers were inserted in the format '[1][2]' at the end of the relevant sentences, and a consolidated "References" list, compiling all unique cited sources, was appended to the conclusion of each article. This uniform approach to citation structure was essential for equitable and rigorous factual verification.

These standardized settings were implemented to minimize variability arising from differing default configurations and to enable a more direct comparison of the models' capabilities in the context of deep research tasks.

# H   DETAILED CALCULATION OF HUMAN CONSISTENCY METRICS

This appendix provides the detailed calculation methods for the four metrics used to validate the consistency between our RACE framework and human judgment, as introduced in Section 4.3.2.

## H.1   PAIRWISE AGREEMENT RATE

The Pairwise Agreement Rate measures the proportion of report pairs (across all tasks) where the evaluation method's preference matches the human preference.

For each of the $N_t = 50$ tasks in our study, four generated deep research reports result in $N_p = \binom{4}{2} = 6$ unique pairs per task. Human preference for each pair (e.g., Report A is better than Report B, or they are tied) is established from the average overall scores assigned to each report by three domain experts.

Let $\mathrm{I}(t, p)$ be an indicator function for task $t$ and pair $p$:

$$\mathrm{I}(t, p) = \begin{cases} 1 & \text{if the method's preference matches the human preference for pair } p \text{ of task } t \\ 0 & \text{otherwise.} \end{cases} \tag{5}$$

The Pairwise Agreement Rate is then calculated as:

$$\text{Pairwise Agreement Rate} = \frac{\sum_{t=1}^{N_t} \sum_{p=1}^{N_p} \mathrm{I}(t, p)}{N_t \times N_p}. \tag{6}$$

This metric reflects the evaluation method's reliability in replicating human comparative judgments.

## H.2   OVERALL PEARSON CORRELATION

This metric quantifies the linear relationship between average model scores from the evaluation method and those from human experts, aggregated across all $N_t = 50$ tasks.

Let $X$ be a vector of average scores per model (e.g., for the different DRAs evaluated) obtained from our method, aggregated across all tasks. Let $Y$ be the corresponding vector of average scores per model obtained from human experts, also aggregated across all tasks. The Overall Pearson Correlation is the standard Pearson correlation coefficient $r(X, Y)$ calculated between these two vectors. This reflects the overall score correlation between the method and human experts for the evaluated DRAs.

## H.3   FILTERED AVERAGE PEARSON CORRELATION

This metric calculates the average of per-task Pearson correlations ($r_t$) between the method's scores and mean human scores, specifically on tasks where human judgment is more consistent.

Given that human scores are from a limited number of experts ($k = 3$) for the $n = 4$ reports per task, expert inconsistencies can affect task-level metric stability. To mitigate this, tasks are filtered based on inter-rater reliability using the Intraclass Correlation Coefficient (ICC). For each task, ICC(1,1) (a one-way random effects model) is computed from the $k = 3$ human experts' scores for the $n = 4$ reports:

$$\mathrm{ICC}(1, 1) = \frac{\mathrm{MSB} - \mathrm{MSW}}{\mathrm{MSB} + (k-1)\mathrm{MSW}}, \tag{7}$$

where MSB is the mean square between reports and MSW is the mean square within reports. Tasks with poor inter-rater reliability (e.g., $\mathrm{ICC}(1, 1) < 0$) are excluded. This yields a filtered subset of $N_{\text{filtered}}$ tasks, denoted as $\mathcal{T}_{\text{filtered}}$ (37 in our experiments).

The Filtered Average Pearson correlation is then the average of per-task Pearson correlations ($r_t$) between the method's scores and mean human scores over $\mathcal{T}_{\text{filtered}}$:

$$\text{Filtered Avg Pearson} = \frac{1}{N_{\text{filtered}}} \sum_{t \in \mathcal{T}_{\text{filtered}}} r_t. \tag{8}$$

This procedure provides a more robust assessment of absolute-score correlation.

## H.4 FILTERED AVERAGE SPEARMAN CORRELATION

Using the same filtering method and the subset $\mathcal{T}_{\text{filtered}}$, this metric evaluates model ranking consistency.

For each task $t \in \mathcal{T}_{\text{filtered}}$, the Spearman rank correlation coefficient $\rho_t$ is calculated between model rankings derived from our evaluation method and those from average human scores. The Filtered Average Spearman Correlation is then the average of these $\rho_t$ values:

$$\text{Filtered Avg Spearman} = \frac{1}{N_{\text{filtered}}} \sum_{t \in \mathcal{T}_{\text{filtered}}} \rho_t. \tag{9}$$

This reflects how well the method preserves relative model ordering compared to humans, specifically on tasks with more consistent human judgments.

# I ANALYSIS OF RACE ROBUSTNESS

## I.1 ROBUSTNESS TO REFERENCE SELECTION

To assess sensitivity to the choice of reference article, we replace the default Gemini-2.5-Pro Deep Research reference with reports from Claude-Research and Kimi-Researcher across the full task set. The ranking remains identical (Gemini > OpenAI > Perplexity > Grok), indicating that RACE is robust to reference selection.

Table 7: RACE results with Claude-Research as reference.

|  | Overall | Comp. | Insight | Inst. | Read. |
|---|---|---|---|---|---|
| Gemini-2.5-Pro Deep Research | **55.30 (1st)** | 55.56 | 56.22 | 52.89 | 56.55 |
| OpenAI Deep Research | **52.27 (2nd)** | 52.70 | 50.73 | 52.80 | 54.41 |
| Perplexity Deep Research | **44.11 (3rd)** | 43.16 | 39.71 | 48.47 | 47.80 |
| Grok Deeper Search | **42.53 (4th)** | 41.29 | 35.96 | 48.89 | 47.29 |

Table 8: RACE results with Kimi-Researcher as reference.

|  | Overall | Comp. | Insight | Inst. | Read. |
|---|---|---|---|---|---|
| Gemini-2.5-Pro Deep Research | **55.22 (1st)** | 55.27 | 57.08 | 52.71 | 55.04 |
| OpenAI Deep Research | **52.38 (2nd)** | 52.48 | 52.21 | 52.49 | 52.91 |
| Perplexity Deep Research | **44.24 (3rd)** | 43.28 | 40.49 | 48.53 | 46.66 |
| Grok Deeper Search | **41.15 (4th)** | 39.31 | 34.64 | 48.13 | 45.72 |

## I.2 ROBUSTNESS TO LENGTH INFLATION

To evaluate robustness to length-based bias, we start from Gemini-2.5-Pro Deep Research articles and segment them into paragraphs. For each paragraph, we prompt `Gemini-2.5-Pro` to rewrite it by expanding length while preserving the original information and maintaining logical coherence with the surrounding context. We perform this process iteratively, yielding average article lengths of approximately $\times 1.47$ (expand) and $\times 2.19$ (expand$^2$) the original. Subsequent RACE evaluations show that scores do not increase with length; under higher expansion they decline, indicating that RACE is resilient to simple length-inflation attacks.

Table 9: Length bias analysis under controlled expansions.

| Target | Avg. Length | Overall | Comp. | Insight | Inst. | Read. |
|---|---|---|---|---|---|---|
| Gemini-DeepResearch | 33.4k | **48.92** | 48.45 | 48.30 | 49.29 | 49.77 |
| Gemini-DeepResearch (expand) | 49.0k ($\times 1.47$) | **48.68** | 48.78 | 48.49 | 49.12 | 47.57 |
| Gemini-DeepResearch (expand$^2$) | 73.2k ($\times 2.19$) | **47.07** | 48.32 | 47.19 | 48.30 | 40.49 |

### I.3 OPEN-SOURCE JUDGE LLM

We additionally evaluated an open-source Judge LLM, `Qwen3-235B-A22B-Thinking-2507`, as a substitute for proprietary models in the RACE framework. The Judge LLM performed the same steps as in our main setup: dynamic dimension weighting, task-specific criteria generation, and reference-based scoring. Hyperparameters and prompts were kept aligned with the Gemini-based setup where applicable.

Using Qwen3 as the Judge, the relative ranking across the four DRAs remains identical to the Gemini-based setup, demonstrating that RACE is robust to the choice of Judge backbone. Compared with closed-source judges, Qwen3 yields lower human-consistency metrics on the correlation scale; however, the gap is moderate, and Qwen3 remains a practical open-source alternative when access or cost constraints arise.

Table 10: Overall RACE scores of four DRAs under different Judge LLMs.

| Judge | Gemini-2.5-Pro DR | OpenAI DR | Perplexity DR | Grok DS |
|---|---|---|---|---|
| Gemini-2.5-pro | 48.88 | 46.98 | 42.25 | 40.24 |
| Qwen3-235b-thinking | 50.17 | 46.88 | 41.25 | 38.04 |

## J ANALYSIS OF RACE GENERALIZABILITY

RewardBench 2 evaluate reward/judge models in generative settings with accuracy-based, multi-skill preference judgments across Factuality, Precise Instruction-Following, Math, Safety, Focus, and handling Ties (Lambert et al., 2024; Malik et al., 2025).

We apply RACE as a general reward-modeling method by using `DeepSeek-R1` as the judge and replacing vanilla direct scoring with RACE's pipeline: (1) dynamically generate evaluation dimensions weights, (2) generate task-specific, executable criteria, and (3) perform reference-based comparative scoring. Other settings follow the RewardBench 2 protocol. With RACE, `DeepSeek-R1` shows substantial gains and approaches leading proprietary judges, indicating that RACE strengthens reward modeling and transfers beyond deep research, consistent with our rebuttal analysis.

Table 11: RewardBench-v2 (Generative) results with and without RACE.

| Model | Score | Factuality | Precise IF | Math | Safety | Focus | Ties |
|---|---|---|---|---|---|---|---|
| LMunit (SOTA) | 82.1 | 87.2 | 54.4 | 72.7 | 91.3 | 96.8 | 90.1 |
| Claude-Opus-4 | 76.5 | 82.7 | 41.9 | 74.9 | 89.5 | 86.2 | 83.7 |
| Gemini-2.5-pro | 74.8 | 71.2 | 52.1 | 68.3 | 88.7 | 79.4 | 81.2 |
| DeepSeek-R1 w/o RACE | 51.5 | 44.4 | 19.9 | 46.2 | 70.1 | 55.5 | 47.7 |
| DeepSeek-R1 w/ RACE | **74.4** | **72.9** | **45.6** | **74.3** | **90.9** | **76.8** | **47.6** |

## K STATISTICAL ANALYSIS OF PERFORMANCE DIFFERENCES

To provide rigorous evidence for the robustness of our evaluation results, we conducted comprehensive statistical analyses examining both the significance of performance differences between agents and the sensitivity of rankings to task selection.

### K.1 PAIRWISE STATISTICAL SIGNIFICANCE

We performed paired $t$-tests between all top-performing DRAs on their common task set. Since one agent (Doubao DR) did not respond to one task, we report results based on 99 tasks common to all agents. Table 12 presents the results.

**Key findings:**

- **Top-tier agents show clear separation**: Gemini-2.5-Pro DR significantly outperforms all others ($p < 0.001$, Cohen's $d$ ranging from 0.65 to 1.19, medium to large effects).

- **OpenAI DR forms a distinct second tier**: It significantly outperforms Claude, Kimi, and Doubao ($p \leq 0.006$, Cohen's $d$ from 0.28 to 0.44), though the effect sizes are smaller than Gemini's advantage.

- **Mid-tier agents are statistically indistinguishable**: Claude Research, Kimi Researcher, and Doubao DR show no significant pairwise differences ($p > 0.18$, Cohen's $d < 0.15$), indicating they perform at a comparable level.

Table 12: Pairwise Statistical Significance Tests (Top-5 DRAs). $*** p < 0.001$; $** p < 0.01$; $* p < 0.05$; n.s. = not significant. $\Delta$: mean difference in percentage points. CI: 95% confidence interval for the difference. Paired $t$-tests based on common tasks.

| Comparison | $n$ | $\Delta$ (%) | 95% CI | $t$-stat | $p$-value | Significance |
|---|---|---|---|---|---|---|
| Gemini-2.5-Pro DR vs OpenAI DR | 100 | 3.26 | [2.27, 4.25] | 6.52 | $< 0.001$ | $***$ |
| Gemini-2.5-Pro DR vs Claude Research | 100 | 4.72 | [3.82, 5.62] | 10.42 | $< 0.001$ | $***$ |
| Gemini-2.5-Pro DR vs Kimi Researcher | 100 | 5.07 | [4.09, 6.06] | 10.24 | $< 0.001$ | $***$ |
| Gemini-2.5-Pro DR vs Doubao DR | 99 | 5.36 | [4.46, 6.26] | 11.81 | $< 0.001$ | $***$ |
| OpenAI DR vs Claude Research | 100 | 1.46 | [0.43, 2.48] | 2.81 | 0.006 | $**$ |
| OpenAI DR vs Kimi Researcher | 100 | 1.81 | [0.91, 2.71] | 4.00 | $< 0.001$ | $***$ |
| OpenAI DR vs Doubao DR | 99 | 2.13 | [1.17, 3.09] | 4.39 | $< 0.001$ | $***$ |
| Claude Research vs Kimi Researcher | 100 | 0.35 | [−0.54, 1.25] | 0.79 | 0.432 | n.s. |
| Claude Research vs Doubao DR | 99 | 0.66 | [−0.33, 1.64] | 1.32 | 0.188 | n.s. |
| Kimi Researcher vs Doubao DR | 99 | 0.28 | [−0.69, 1.26] | 0.58 | 0.564 | n.s. |

These results confirm that the observed performance gaps in Table 1 are **statistically robust**, not artifacts of random variation. The presence of both significant and non-significant differences demonstrates that our evaluation can discriminate meaningful performance gaps while correctly identifying agents of similar capability.

## K.2 TASK SAMPLING SENSITIVITY

To assess whether rankings depend on the specific 99–100 tasks selected, we performed leave-one-out (LOO) analysis: for each task, we removed it and recomputed rankings on the remaining tasks.

Table 13 shows that all agents maintained their ranks in **100% of LOO iterations**. This perfect stability indicates that rankings are not driven by any single critical task, performance differences are consistent across task subsets, and the benchmark size (99–100 tasks) is sufficient to produce stable rankings.

Table 13: Leave-One-Out Ranking Stability Analysis. Based on 99 tasks common to all evaluated agents. Consistency: percentage of LOO iterations where the agent maintained its rank.

| Position | Agent | Consistency (%) |
|---|---|---|
| 1 | Gemini-2.5-Pro Deep Research | 100.0 |
| 2 | OpenAI Deep Research | 100.0 |
| 3 | Claude Research | 100.0 |
| 4 | Kimi Researcher | 100.0 |
| 5 | Doubao Deep Research | 100.0 |

## K.3 DISCUSSION

The statistical analyses presented above establish the robustness of our evaluation framework from two complementary perspectives:

**Performance differentiation:** Pairwise comparisons reveal a clear performance hierarchy with statistically significant gaps between top performers and others. Crucially, we also observe non-significant differences among mid-tier agents, demonstrating that our framework appropriately identifies both meaningful differences and statistical ties.

**Ranking stability:** Perfect LOO stability shows that rankings are highly robust to task selection. The 99–100 task scale provides sufficient statistical power to produce reliable, generalizable results.

Together, these analyses validate that DeepResearch Bench produces **statistically sound and stable** evaluations, suitable for drawing meaningful conclusions about DRA capabilities.

## L  ADDITIONAL DEEP RESEARCH TASK EXAMPLES

- "Investigate how, under chronic antigen stimulation (e.g., the tumor micro-environment or latent HIV infection), mitochondrial dynamics (fusion–fission balance) in $CD8^+$ T cells drive bifurcation into terminally exhausted and tissue-resident memory (Trm) fates via epigenetic reprogramming (e.g., m6A modification, lactate-mediated histone lactylation). Develop quantitative models based on metabolic–epigenetic interaction networks."

- "Analyze liability allocation in accidents involving vehicles with advanced driver-assistance systems (ADAS) operating in a shared human–machine driving context. Integrate technical principles of ADAS, existing legal frameworks, and relevant case law to systematically examine the boundaries of responsibility between the driver and the system."

- "How can we conduct comprehensive and accurate situational awareness of space targets in cislunar space, and support the effectiveness of short-term cislunar tracking and monitoring tasks? Compare existing sensing architectures, data-fusion pipelines, and control strategies."

## M    PROMPT TEMPLATES

---

**Clean Article Prompt**

```
<system_role>
```
You are a professional article editor who is good at cleaning and refining article content.
```
</system_role>
<user_prompt>
```
Please help me clean the following research article, removing all citation links, citation marks (such as [1], [2], 1, 2, etc. or other complex citation formats), reference lists, footnotes, and ensuring the content is coherent and smooth. Keep all other original content of the article, removing only the citations. If the content of the citation mark is used as part of a sentence in the article, keep the text content and remove other marks.
Article content: "{article}"
Please return the cleaned article in full, without adding any additional comments or explanations.
```
</user_prompt>
```

---

**Generate Dynamic Dimension Weight Prompt**

```
<system_role>
```
You are an experienced research article evaluation expert. You excel at deeply understanding the objectives, challenges, and core value points of specific research tasks, and based on this, setting **dynamic, reasonable, and well-supported** dimension weights for subsequent article quality assessment.
```
</system_role>
<user_prompt>
```
There is a deep research task as follows:
```
<task>
```
"{task_prompt}"
```
</task>
<instruction>
```
**Background**: The research team will conduct in-depth and comprehensive research based on the '`<task>`' above and ultimately produce a high-quality research article.

**Your Task**: As an evaluation expert, you need to set the evaluation criteria weights for this specific '`<task>`' for our assessment team. The evaluation will be conducted across the following four dimensions:

1. **Comprehensiveness:** The breadth, depth, and relevance of information coverage.

2. **Insight:** The depth, originality, logic, and value of the analysis and conclusions.

3. **Instruction Following:** Whether the report accurately and completely responds to all requirements and constraints of the task.

4. **Readability:** Clarity of structure, fluency of language, effectiveness of data presentation, and overall ease of understanding.

**Evaluation Formula**: Total Score = Comprehensiveness * Comprehensiveness Weight + Insight * Insight Weight + Instruction Following * Instruction Following Weight + Readability * Readability Weight.
(**Note: The sum of all weights must be exactly 1.0**)
**Core Requirements**:

1. **In-depth Task Analysis**: Carefully study the specific content of the '`<task>`', its implicit goals, potential difficulties, and the core value of its outcomes.

2. **Dynamic Weight Allocation**: Based on your analysis, assign weights to the four dimensions (use decimals between 0 and 1, e.g., 0.3).

   **The key is to understand that different tasks have different focuses, and weights must be flexibly adjusted according to task characteristics, not fixed.**

3. **Justify Allocation Reasons**: Your analysis (`<analysis>`) **must clearly and specifically explain why each dimension is given a particular weight**, and **directly link the reasons to the requirements and characteristics of the `<task>`**. This is crucial for evaluating the quality of your work.

4. **Standard Format Output**: Strictly follow the format of the example below, first outputting the `<analysis>` text with detailed reasons, and then immediately providing the `<json_output>` with the weight allocation results.

```
</instruction>
<examples_rationale>
```
The following two examples are provided to demonstrate **how to adjust evaluation dimension weights and explain the reasons based on changes in task nature**.
Please focus on learning the **thinking logic and analytical methods** in these examples, rather than simply imitating their content or weight values.
```
</examples_rationale>
<example_1>
<task>
```
"Analyze the feasibility of investing in electric vehicle (EV) charging infrastructure in suburban areas."
```
</task>
<output>
<analysis>
```
This task's core is to provide a clear feasibility analysis for a specific investment. The value lies in the thoroughness of the assessment and the practicality of its conclusions. Therefore, evaluation emphasizes insight and comprehensiveness.

- **Insight (0.35):** The task requires a deep analysis of feasibility. The quality of the strategic recommendations derived from this analysis is key.

- **Comprehensiveness (0.30):** A thorough investigation of all relevant factors (technical, economic, social, environmental) is crucial for a reliable feasibility study.

- **Instruction Following (0.20):** The report must specifically address EV charging infrastructure in suburban areas and focus on investment feasibility.

- **Readability (0.15):** Clearly communicating complex financial and technical analysis is important, but secondary to the depth and breadth of the study.

```
</analysis>
<json_output>
```
{{ "comprehensiveness": 0.30, "insight": 0.35, "instruction_following": 0.20, "readability": 0.15 }}
```
</json_output>
</output>
</example_1>
```
Please strictly follow the above instructions and methods. Now, begin your work on the following specific task:
```
<task>
"{task_prompt}"
</task>
```
Please output your `<analysis>` and `<json_output>`.
```
</user_prompt>
```

**Generate Comprehensiveness Criteria Prompt**

```
<system_role>
```
You are an experienced research article evaluation expert. You excel at breaking down abstract evaluation dimensions (like "Comprehensiveness") into actionable, clear, and task-specific criteria, assigning appropriate weights and justifications for each.
```
</system_role>
<user_prompt>
```
**Background**: We are evaluating a deep research article written for the following task across four dimensions: Comprehensiveness, Insight, Instruction Following, and Readability.

1. **Comprehensiveness:** The breadth, depth, and relevance of information coverage.

2. **Insight:** The depth, originality, logic, and value of the analysis and conclusions.

3. **Instruction Following:** Whether the report accurately and completely responds to all requirements and constraints of the task.

4. **Readability:** Clarity of structure, fluency of language, effectiveness of data presentation, and overall ease of understanding.

```
<task>
"{task_prompt}"
</task>
<instruction>
```
**Your Goal**: For the **Comprehensiveness** dimension of this research article, develop a set of detailed, specific, and highly task-relevant evaluation criteria. You need to:

1. **Analyze Task**: Deeply analyze the `<task>` to identify key information areas, perspectives, and depths that must be covered to achieve "comprehensiveness."

2. **Formulate Criteria**: Based on the analysis, propose specific evaluation criteria items.

3. **Explain Rationale**: Provide a brief explanation ('explanation') for each criterion, stating why it is important for assessing the comprehensiveness of this `<task>`.

4. ...

**Core Requirements**:

1. **Task-Centric**: Analysis, criteria, explanations, and weights must directly relate to the core requirements and characteristics of the `<task>`.

2. **Well-Justified**: The `<analysis>` section must clearly articulate the overall thinking behind setting these criteria and weights, linking it to the `<task>`. The 'explanation' for each criterion must justify its specific relevance.

3. ...

```
</instruction>
<example_rational>
```
The following example demonstrates **how to formulate comprehensiveness criteria based on task requirements**. Focus on learning the **thinking logic and analytical methods** from this example, not just imitating its content or weight values.
```
</example_rational>
<example_1>
<task>
```
"Analyze the impact of remote work trends on commercial real estate in major US cities and recommend investment strategies."
```
</task>
<output>
<analysis>
```
To comprehensively evaluate a research article on "the impact of remote work on commercial real estate in major US cities and recommended investment strategies," considerations must span multiple dimensions.

Specifically, evaluation criteria need to cover:

1. **Remote Work Trends & Adoption Data**: Coverage of current and projected remote/hybrid work models, adoption rates across industries and demographics.

2. **Impact on Commercial Real Estate Sectors**: Analysis of effects on office, retail, and industrial spaces, including vacancy rates, leasing trends, and property valuations in major US cities.

3. **Geographical Variations**: Examination of how impacts differ across various major US cities (e.g., tech hubs vs. financial centers, downtown vs. suburban).

4. ...

Weight allocation should be balanced between the impact analysis...
```
</analysis>
<json_output>
[
{{
"criterion": "Analysis of Remote Work Trends and Adoption",
"explanation": "Assesses if the article thoroughly examines current and projected remote/hybrid work models...",
"weight": 0.15
}},
{{
"criterion": "Comprehensive Coverage of CRE Sector Impacts",
"explanation": "...",
"weight": 0.20
}},
{{
"criterion": "Examination of Geographical Variations and Nuances",
"explanation": "...",
"weight": 0.15
}},
{{
"criterion": "Discussion of Broader Economic and Social Consequences",
"explanation": "...",
"weight": 0.10
}},
...
]
</json_output>
</output>
</example_1>
```
Please strictly follow the above instructions and methods. Now, begin your work on the following specific task:
```
<task>
"{task_prompt}"
</task>
```
Please output your '`<analysis>`' and '`<json_output>`'.
```
</user_prompt>
```

---

**Score Prompt In RACE(Full)**

```
<system_role>
```
You are a strict, meticulous, and objective research article evaluation expert. You excel at using specific assessment criteria to deeply compare two articles on the same task, providing precise scores and clear justifications.
```
</system_role>
<user_prompt>
```
**Task Background**

There is a deep research task, and you need to evaluate two research articles written for this task. We will assess the articles across four dimensions: Comprehensiveness, Insight, Instruction Following, and Readability. The content is as follows:
```
<task>
```
"{task_prompt}"
```
</task>
```
**Articles to Evaluate**
```
<article_1>
```
"{article_1}"
```
</article_1>
<article_2>
```
"{article_2}"
```
</article_2>
```
**Evaluation Criteria** Now, you need to evaluate and compare these two articles based on the following **evaluation criteria list**, providing comparative analysis and scoring each on a scale of 0-10. Each criterion includes an explanation, please understand carefully.
```
<criteria_list>
```
{criteria_list}
```
</criteria_list>
<Instruction>
```
**Your Task**

Please strictly evaluate and compare '`<article_1>`' and '`<article_2>`' based on **each criterion** in the '`<criteria_list>`'. You need to:

1. **Analyze Each Criterion**: Consider how each article fulfills the requirements of each criterion.

2. **Comparative Evaluation**: Analyze how the two articles perform on each criterion, referencing the content and criterion explanation.

3. **Score Separately**: Based on your comparative analysis, score each article on each criterion (0-10 points).

**Scoring Rules**

For each criterion, score both articles on a scale of 0-10 (continuous values). The score should reflect the quality of performance on that criterion:

- 0-2 points: Very poor performance. Almost completely fails to meet the criterion requirements.

- 2-4 points: Poor performance. Minimally meets the criterion requirements with significant deficiencies.

- 4-6 points: Average performance. Basically meets the criterion requirements, neither good nor bad.

- 6-8 points: Good performance. Largely meets the criterion requirements with notable strengths.

- 8-10 points: Excellent/outstanding performance. Fully meets or exceeds the criterion requirements.

**Output Format Requirements**

Please **strictly** follow the '`<output_format>`' below for each criterion evaluation. **Do not include any other unrelated content, introduction, or summary**. Start with "Standard 1" and proceed sequentially through all criteria:

```
</Instruction>
<output_format>
```
{{
"comprehensiveness": [
{{
"criterion": [Text content of the first comprehensiveness evaluation criterion],
"analysis": [Comparative analysis],
"article_1_score": [Continuous score 0-10],
"article_2_score": [Continuous score 0-10]
}},
{{
"criterion": [Text content of the second comprehensiveness evaluation criterion],
"analysis": [Comparative analysis],
"article_1_score": [Continuous score 0-10],
"article_2_score": [Continuous score 0-10]
}},
...
],
"insight": [
{{
"criterion": [Text content of the first insight evaluation criterion],
"analysis": [Comparative analysis],
"article_1_score": [Continuous score 0-10],
"article_2_score": [Continuous score 0-10]
}},
...
],
...
}}
```
</output_format>
```
Now, please evaluate the two articles based on the research task and criteria, providing detailed comparative analysis and scores according to the requirements above. Ensure your output follows the specified '`<output_format>`' and that the JSON format is parsable, with all characters that might cause JSON parsing errors properly escaped.
```
</user_prompt>
```

**Static Score Prompt**

```
<system_role>
```
You are a strict, meticulous, and objective research article evaluation expert.
You excel at using specific assessment criteria to deeply compare two articles on the same task, providing precise scores and clear justifications.
```
</system_role>
<user_prompt>
```
**Task Background**
There is a deep research task, and you need to evaluate two research articles written for this task.
We will assess the articles across four dimensions: Comprehensiveness, Insight, Instruction Following, and Readability.
The content is as follows:
```
<task>
```
"{task_prompt}"
```
</task>
```
**Articles to Evaluate**
```
<article_1>
```
"{article_1}"
```
</article_1>
<article_2>
```
"{article_2}"
```
</article_2>
```
**Evaluation Criteria**
Now, you need to evaluate and compare these two articles based on the following **fixed evaluation criteria list**, providing comparative analysis and scoring each on a scale of 0-10. Each criterion includes an explanation, please understand carefully.
```
<criteria_list>
```
# Comprehensiveness
[ {{ "criterion": "Information Coverage Breadth",
"explanation": "Evaluates whether the article covers all key areas and aspects related to the topic without omitting important information.",
"weight": 0.25
}},
{{
"criterion": "Information Depth and Detail",
"explanation": "...",
"weight": 0.25
}},
{{ "criterion": "Data and Factual Support",
"explanation": "...",
"weight": 0.25
}},
{{
"criterion": "Multiple Perspectives and Balance",
"explanation": "...",
"weight": 0.25
}} ]
# Insight
[ {{ "criterion": "Analysis Depth and Originality",
"explanation": "...",
"weight": 0.25
}},
... ]
...
```
</criteria_list>
<Instruction>
```

**Your Task**
Please strictly evaluate and compare '`<article_1>`' and '`<article_2>`' based on **each criterion** in the '`<criteria_list>`'.
You need to:

1. **Analyze Each Criterion**: Consider how each article fulfills the requirements of each criterion.

2. **Comparative Evaluation**: Analyze how the two articles perform on each criterion, referencing the content and criterion explanation.

3. **Score Separately**: Based on your comparative analysis, score each article on each criterion (0-10 points).

**Scoring Rules**
For each criterion, score both articles on a scale of 0-10 (continuous values).
The score should reflect the quality of performance on that criterion:

- 0-2 points: Very poor performance. Almost completely fails to meet the criterion requirements.

- ...

- 8-10 points: Excellent/outstanding performance. Fully meets or exceeds the criterion requirements.

**Output Format Requirements**
Please **strictly** follow the '`<output_format>`' below for each criterion evaluation.
**Do not include any other unrelated content, introduction, or summary**.
Start with "Standard 1" and proceed sequentially through all criteria:
```
</Instruction>
<output_format_comp>
```
{{
"comprehensiveness": [
{{
"criterion": [Text content of the first comprehensiveness evaluation criterion],
"analysis": [Comparative analysis],
"article_1_score": [Continuous score 0-10],
"article_2_score": [Continuous score 0-10]
}},
{{
...
}},
...
], "insight": [
{{
"criterion": [Text content of the first insight evaluation criterion],
"analysis": [Comparative analysis],
"article_1_score": [Continuous score 0-10],
"article_2_score": [Continuous score 0-10]
}},
...
],
...
}}
```
</output_format>
```
Now, please evaluate the two articles based on the research task and criteria, providing detailed comparative analysis and scores according to the requirements above.
Ensure your output follows the specified '`<output_format>`' and that the JSON format is parsable, with all characters that might cause JSON parsing errors properly escaped.
```
</user_prompt>
```

---

**Point-wise Score Prompt**

```
<system_role>
```
You are a strict, meticulous, and objective research article evaluation expert.
You excel at using specific assessment criteria to thoroughly evaluate research articles, providing precise scores and clear justifications.
```
</system_role>
<user_prompt>
```
**Task Background**
There is a deep research task, and you need to evaluate a research article written for this task.
We will assess the article across four dimensions: Comprehensiveness, Insight, Instruction Following, and Readability.
The content is as follows:
```
<task>
```
"{task_prompt}"
```
</task>
```
**Article to Evaluate**
```
<target_article>
```
"{article}"
```
</target_article>
```
**Evaluation Criteria**
Now, you need to evaluate this article based on the following **evaluation criteria list**, providing analysis and scoring each on a scale of 0-10.
Each criterion includes an explanation, please understand carefully.
```
<criteria_list>
```
{criteria_list}
```
</criteria_list>
<Instruction>
```
**Your Task**
Please strictly evaluate `<target_article>` based on **each criterion** in the `<criteria_list>`.
You need to:

1. **Analyze Each Criterion**: Consider how the article fulfills the requirements of each criterion.

2. **Analysis and Evaluation**: Analyze the article's performance on each criterion, referencing the content and criterion explanation, noting strengths and weaknesses.

3. **Score**: Based on your analysis, score the article on each criterion (0-10 points).

**Scoring Rules**
For each criterion, score the article on a scale of 0-10 (continuous values).
The score should reflect the quality of performance on that criterion:

- 0-2 points: Very poor performance. Almost completely fails to meet the criterion requirements.

- 2-4 points: Poor performance. Minimally meets the criterion requirements with significant deficiencies.

- 4-6 points: Average performance. Basically meets the criterion requirements, neither good nor bad.

- 6-8 points: Good performance. Largely meets the criterion requirements with notable strengths.

- 8-10 points: Excellent/outstanding performance. Fully meets or exceeds the criterion requirements.

**Output Format Requirements**
Please **strictly** follow the `<output_format>` below for each criterion evaluation.
**Do not include any other unrelated content, introduction, or summary**.
Start with "Standard 1" and proceed sequentially through all criteria:

```
</Instruction>
<output_format>
{{
"comprehensiveness": [
{{
"criterion": [Text content of the first comprehensiveness evaluation criterion],
"analysis": [Analysis],
"target_score": [Continuous score 0-10]
}},
{{
"criterion": [Text content of the second comprehensiveness evaluation criterion],
"analysis": [Analysis],
"target_score": [Continuous score 0-10]
}},
...
],
"insight": [
{{
"criterion": [Text content of the first insight evaluation criterion],
"analysis": [Analysis],
"target_score": [Continuous score 0-10]
}},
...
],
...
}}
</output_format>
```
Now, please evaluate the article based on the research task and criteria, providing detailed analysis and scores according to the requirements above.
Ensure your output follows the specified '`<output_format>`' and that the JSON format is parsable, with all characters that might cause JSON parsing errors properly escaped.
```
</user_prompt>
```

**Vanilla Prompt**

```
<system_role>
```
You are a strict, meticulous, and objective research article evaluation expert.
You excel at using specific assessment criteria to thoroughly evaluate research articles, providing precise scores and clear justifications.
```
</system_role>
<user_prompt>
```
**Task Background**
There is a deep research task, and you need to evaluate a research article written for this task.
```
<task>
```
"{task_prompt}"
```
</task>
```
**Article to Evaluate**
```
<target_article>
```
"{article}"
```
</target_article>
<Instruction>
```
**Your Task**
Please evaluate the overall quality of the above '`<target_article>`' as a response to '`<task>`'.
Please provide an overall score between 0 and 10.
Also, provide a brief justification for your score.
**Output Format Requirements**
Please **strictly** follow the '<output_format>' below for your evaluation result.
**Do not include any other unrelated content, introduction, or summary**.
```
</Instruction>
<output_format>
```
{{ "overall_score": [Continuous score 0-10], "justification": "[Scoring justification]" }}
```
</output_format>
```
Now, please evaluate the article based on the task and provide your score and justification according to the specified format.
Ensure your output is valid JSON format and escape any special characters as needed.
```
</user_prompt>
```

