# OpenReview forum: "DeepResearch Bench: A Comprehensive Benchmark for Deep Research Agents"
_ICLR.cc/2026/Conference — ICLR 2026 Poster_

### Official Review · Reviewer_8pm9 · 2025-10-26

**Soundness:** 4
**Presentation:** 4
**Contribution:** 4
**Rating:** 8
**Confidence:** 4

**Summary:**

The paper presents DeepResearch Bench, a benchmark for evaluating Deep Research Agents (DRAs) that perform open-ended web-based research. It comprises 100 expert-curated tasks across 22 domains and introduces two automated evaluation frameworks: RACE for report quality and FACT for retrieval accuracy. Experiments and human studies show strong alignment between these frameworks and expert judgments. Overall, the work provides a solid foundation for systematic and scalable evaluation of LLM-based research agents.

**Strengths:**

1.	The paper introduces a well-motivated and comprehensive benchmark for Deep Research Agents, grounded in real-world user queries and expert-curated tasks across 22 domains, resulting in a high-quality and realistic evaluation suite.
2.	The proposed RACE and FACT frameworks jointly assess report quality and retrieval reliability, effectively addressing a key gap in existing evaluation methods — making this a timely and impactful contribution.
3.	The experiments are extensive and insightful, covering both commercial and open-source DRAs, and offering clear comparative results and valuable guidance for future research and development.

**Weaknesses:**

There is no major weakness in this paper. The only minor concern lies in the size of the benchmark — with 100 tasks, its scale may still limit robustness for fine-grained statistical analysis or long-tail domain evaluation. However, given the high quality and expert-level design of the tasks, this limitation is acceptable and does not detract from the overall contribution.

**Questions:**

n/a

---

> ### Author Response · Authors · 2025-11-22
>
> We sincerely thank the reviewer for the highly encouraging feedback and positive assessment of our work. Your recognition and support motivate us to continue improving and expanding this benchmark for the research community.

---

### Official Review · Reviewer_oeAc · 2025-10-30

**Soundness:** 2
**Presentation:** 2
**Contribution:** 2
**Rating:** 4
**Confidence:** 3

**Summary:**

In order to evaluate the performance of Deep Research Agents (DRAs) that gather and summarize information for research tasks, this paper presents a benchmark named DeepResearch Bench, with 100 PhD-level research tasks. This work also proposes two automatic methods for evaluation with Judge LLM, including a reference-based approach to measure report quality using adaptive criteria and reference reports, and a citation-based approach assessing the accuracy and effectiveness of retrieved sources. This paper further shows its consistency with human evaluation to highlight its potential.

**Strengths:**

1. This paper contributes the first benchmark to evaluate Deep Research Agents. The design is also grounded in the real-world human research tasks, which are collected based on user interaction with Chatbots.

2. The work conducted comprehensive experiments on popular DRAs, including one open-sourced one, and also compared the DRAs with search-enabled LLMs.

3. The work also analyzed the consistency of the proposed evaluation with human evaluation.

**Weaknesses:**

1. The major concern comes from the validation of the consistency with human evaluation. It's unclear why the final set of tasks is designed to have 50 tasks for Chinese and English, respectively, and which split the experiments are conducted on. It's noted in line 364 that the human evaluation was conducted on the Chinese tasks, which might limit the claim of consistency with human evaluation if the main results are obtained on English tasks.

2. There are 100 tasks in total, with 50 Chinese and 50 English tasks, across 22 domains. The number of tasks per domain and language might be limited.

3. The authors conducted a comparison of human evaluation consistency using various Judge LLMs to select the candidate for Judge LLM. However, the performance of the proposed metrics is highly dependent on the Judge LLM, and the RACE evaluation also relies on the reference reports. While Gemini models are used as the Judge LLM, and Gemini-generated reports are used as the reference reports, this might introduce model bias and favor Gemini-based DRAs, which is also shown in the experiment results.

4. The introduction of the evaluation frameworks in Section 3 is fragmented and eliminates many details, such as the computation of the FACT framework. More complete descriptions of the methods should be included in the main text to increase clarity, potentially moving space from Figures 1, 3, and 4.

**Questions:**

1. How is the variance across the domain or across tasks within a domain?

2. As the final set includes English and Chinese tasks, if both are evaluated for the main results, is there any insight into the DRA performance across languages?

---

> ### Author Response · Authors · 2025-11-22
>
> We thank the reviewer for the encouraging feedback and constructive suggestions.
>
> # Weaknesses
>
> ### ●  **To W1: Human Consistency Validation and Task Split**
>
> We appreciate the reviewer raising this important clarification. To address the concerns:
>
> **Main results are evaluated on all 100 tasks.** As stated throughout Section 4 and Table, all reported performance metrics are computed over the complete set of 100 tasks (50 Chinese + 50 English).
>
> **Why 50 Chinese + 50 English?** We designed the benchmark with bilingual coverage because leading frontier benchmarks increasingly emerge both Chinese and English versions (e.g., BrowseComp and BrowseComp-zh). This design ensures our benchmark reflects the multilingual nature of real-world deep research demands.
>
> **Why human consistency experiments on Chinese tasks only?** As discussed in Section 5 (Limitations), expert judging is costly and limits sample sizes. Given resource constraints, we prioritized maximizing domain coverage (all 50 Chinese tasks across all 22 domains) by recruiting Chinese native-speaker domain experts, rather than recruiting bilingual experts which would have significantly increased cost and reduced coverage breadth.
>
> We acknowledge that these design choices and their rationale were insufficiently explained in the original submission. Following the reviewer's suggestion, we have added clarifications in the **revised paper** (highlighted in blue): Section 2.2 clarifies the bilingual design rationale, and Section 4.1 (Experimental Setup) explicitly states that all main results are evaluated on the complete 100-task set.
>
> ### ●  **To W2: Limited Task Number per Domain and Language**
>
> We acknowledge that the limited number of tasks per domain and language constrains fine-grained within-domain analysis. However, such analysis is **not our primary goal**. Our focus is **end-to-end evaluation of overall DRA capabilities** across diverse research scenarios. The current distribution (100 tasks across 22 domains with bilingual coverage) provides sufficient breadth to assess general-purpose deep research agents' performance across varied topics and languages. We hope this work inspires domain-specific benchmarks for emerging sub-fields where deeper, within-domain focus is needed.
>
> ### ●  **To W3: Model Bias from Using Gemini as Both Judge and Reference**
>
> We respectfully clarify that **Gemini achieving the highest score is not due to same-model bias**. In our human consistency experiments (Section 4.3.1), domain experts also rated Gemini-generated reports highest, consistent with our automated evaluation rankings. The strong correlation between RACE scores and human judgments (Section 4.3.2) confirms that the ranking reflects genuine quality differences, not evaluation bias.
>
> Regarding the design rationale: Our RACE framework requires a high-quality reference report for each task. Inviting domain experts to produce 100 research reports is impractical and unscalable. Therefore, we must select the highest-quality DRA from available systems. We chose **Gemini-2.5-Pro Deep Research** for its quality, transparency, and accessibility (Section 4.1).
>
> To address bias concerns, we conducted **comprehensive robustness experiments** (Section 4.3 and Appendix "ANALYSIS OF RACE ROBUSTNESS"). These experiments demonstrate that the RACE framework's design effectively **minimizes same-model bias impact**: Changing the reference or judge does not affect the final conclusion (rankings remain stable).
>
> These results confirm that our evaluation outcomes are robust and not artifacts of using Gemini as both judge and reference.
>
> ### ●  **To W4: Fragmented Introduction of Evaluation Frameworks**
>
> We sincerely appreciate this valuable suggestion. When preparing the initial submission, we placed many computational details in the Appendix due to page limit constraints. Fortunately, ICLR allows 10 pages for main content during the rebuttal phase. We have adjusted the paper structure to include more complete descriptions in the main text. Specifically, we have integrated the detailed calculation methods for the FACT framework's Citation Accuracy and Effective Citations metrics directly into Section 3.2 (**highlighted in blue in the revised paper**), and removed the corresponding appendix section. This should significantly improve clarity for readers.

---

> > ### Author Response · Authors · 2025-11-22
> >
> > # Questions
> >
> > ### ●  **To Q1: Variance Across Domains**
> >
> > Due to the limited number of tasks per domain (2-4 tasks in several domains due to real-world distribution), within-domain variance analysis would not be meaningful. Instead, we present the **cross-domain performance** of each DRA to show which models demonstrate better stability across different research domains.
> >
> > **Cross-Domain Stability Analysis:** We compute the standard deviation of each DRA's performance across 22 domains as a measure of stability. Lower standard deviation indicates more consistent performance across different research areas.
> >
> > | DRA | Cross-Domain Std | Mean Score |
> > |-----|------------------|------------|
> > | LangChain(GPT-4.1) | 0.0147 | 0.438 |
> > | OpenAI | 0.0169 | 0.461 |
> > | Doubao | 0.0176 | 0.441 |
> > | Grok | 0.0187 | 0.388 |
> > | Gemini | 0.0201 | 0.495 |
> > | Perplexity | 0.0207 | 0.405 |
> > | Claude | 0.0222 | 0.453 |
> > | Kimi | 0.0274 | 0.443 |
> >
> > **Key Findings:**
> > - **Most stable**: LangChain (Std=0.0147), showing the most consistent performance across research domains
> > - **Least stable**: Kimi (Std=0.0274), with nearly 2× higher variance
> > - This suggests that while all DRAs can handle diverse domains, their robustness varies significantly
> >
> > ### ●  **To Q2: Performance Across Languages**
> >
> > We compare DRA performance on English versus Chinese tasks. Note that Δ = EN - ZH (positive values indicate better English performance).
> >
> > | Metric | Claude | Doubao | Gemini | Grok | Kimi | LangChain | OpenAI | Perplexity | Overall |
> > |---|---|---|---|---|---|---|---|---|---|
> > | English | 0.461 | 0.447 | 0.499 | 0.391 | 0.457 | 0.439 | 0.470 | 0.417 | 0.448 |
> > | Chinese | 0.439 | 0.440 | 0.496 | 0.374 | 0.436 | 0.430 | 0.459 | 0.392 | 0.433 |
> > | Δ (EN-ZH) | 0.023 | 0.007 | 0.003 | 0.017 | 0.020 | 0.009 | 0.011 | 0.025 | 0.014 |
> >
> > **Key Findings:**
> > - **All DRAs perform better on English tasks**, with an overall average improvement of 0.014 points (on a 0-1 scale)
> > - 3 out of 8 DRAs show statistically significant differences (p<0.05): Claude, Kimi, and Perplexity
> > - **Largest gap**: Perplexity (0.025 points); **Smallest gap**: Gemini (0.003 points)
> >
> > This systematic English advantage suggests that most DRAs' underlying LLMs are **trained more extensively on English corpora**. It is also possible that the reference reports (generated by Gemini-2.5-Pro Deep Research) may have quality differences across languages. However, Gemini itself shows minimal cross-language performance gap (0.003, nearly negligible), suggesting the reference quality difference is not the primary factor. The English advantage more likely reflects the inherent capabilities of each DRA's underlying LLM.

---

### Official Review · Reviewer_xj7Q · 2025-10-31

**Soundness:** 3
**Presentation:** 3
**Contribution:** 3
**Rating:** 6
**Confidence:** 4

**Summary:**

This paper introduces DeepResearch Bench, a benchmark designed to evaluate deep research systems. The benchmark consists of 100 expert-authored PhD-level tasks across 22 domains. For evaluation, the authors propose two methods i.e.,
(1) RACE which uses LLM-as-a-Judge across four dimensions (comprehensiveness, depth, instruction-following, and readability),
(2) FACT, which assesses citation accuracy and the number of verifiable evidence sources.
They ran experiments comparing both deep research agents and llm + web tool calls. they also conducted human eval to study alignment between llm as a judge and expert human annotations.

**Strengths:**

- The problem is timely and there is not many standard benchmarks nor evaluation strategies  designed for deep research systems.
- The problems encompass a large set of topics.
- they evaluate from several different perspectives i.e., both quality of the report as well as citation quality.
- Conducted human eval to validate llm based evaluation
- covered deep research systems as well as llm + search tools

**Weaknesses:**

- Some formatting problems in prompt template Appendix M
- The baseline set omits several relevant open-source and hybrid systems such as Deep Researcher, OpenScholar, or opensrouced LLM + tool-use/web-agent frameworks. T
- It is not explained how the 100 benchmark tasks were chosen from 44 K filtered queries. Did annotators manually reviewed all queries to come up with 100 queries?
-  The benchmark lacks any quantitative or qualitative analysis of task difficulty.
- In Section 3.1.1, it is unclear whether the RACE criteria are fixed per task or dynamically generated per query.
- The FACT framework measures accuracy and count but lacks a notion of coverage i.e., how many claims in the report remain unsupported. Please see https://arxiv.org/abs/2411.17375 for claim coverage eval.
- Section 4.1.1 implies that Gemini-generated reports serve as the reference. This design choice risks biasing evaluation in favor of Gemini-based agents.

**Questions:**

- How were the final 100 benchmark tasks selected from the 40k+ samples?
- Are the task difficulty levels annotated or estimated in any way? How is the distribution of difficulty across the 40k samples and 100 selected questions?
- Are RACE’s adaptive criteria regenerated for each query instance or fixed once per domain?
- Was the Judge LLM calibrated  before correlation tests?
- Why were Gemini-generated reports used as reference as mentioned in 4.1.1 ? could this bias results toward Gemini agents?

---

> ### Author Response · Authors · 2025-11-22
>
> We thank the reviewer for the encouraging feedback and helpful suggestions.
>
> # Weakness & Questions
>
> ### ●  **To W1: Formatting Problems in Prompt Templates**
>
> We thank the reviewer for the careful review. We have fixed the formatting issues in Appendix section "Prompt Templates".
>
> ### ●  **To W2: Baseline Coverage**
>
> We would like to clarify that at the time of this work, there were **no suitable open-source baselines that fit the deep research setting.** Specifically: (1) Early open-source works like DeepResearcher primarily focused on end-to-end RL training for QA-style tasks. In practical deployment, these systems often fail to follow instructions for generating long-form reports required by deep research tasks. (2) OpenScholar restricts retrieval to specific domains and employs offline search after retrieval, which does not align with the interactive, multi-turn search characteristic of deep research agents.
>
> Nevertheless, we greatly appreciate the reviewer's suggestion. To demonstrate the gap between early open-source deep research work and proprietary agents, **we have evaluated DeepResearcher in the revised paper. Additionally, we include Tongyi DeepResearch, the most recent and influential open-source DRA work.**
>
> Key findings: (1) Early RL-based work trained on QA tasks exhibits overfitting behavior—DeepResearcher (10.77) fails to generate complete long-form reports on DRBench tasks. (2) Tongyi DeepResearch demonstrates remarkable robustness: despite its technical report not mentioning specialized training for long-form writing, it achieves performance (40.46) comparable to proprietary agents such as Grok Deeper Search and Perplexity Deep Research.
>
> ### ●  **To W3 & Q1: Task Selection Process**
>
> We clarify that this is described in Section 2.2 and Section 2.1. The 44K samples were used to **analyze the real-world distribution of deep research tasks**, not as a pool for selecting the 100 benchmark tasks. The 100 tasks were **crafted by PhD-level domain experts**, not selected from the 44K samples.
>
> Specifically, the distribution derived from the 44K samples determined the final testset's task allocation—for example, if 12% of the 44K queries were finance-related, our testset would include 12 finance-related deep research tasks, which were then **independently created by finance domain experts**. This approach ensures both authenticity (distribution reflects real demand) and quality (tasks are expert-crafted). To avoid confusion, we have also added a clarification sentence in the **revised paper** (highlighted in blue).
>
> ### ●  **To W4: Task Difficulty Analysis**
>
> We appreciate this concern. We acknowledge that quantifying research task difficulty is inherently subjective, as it depends on domain expertise, information accessibility, and problem complexity. To ensure appropriate difficulty levels, we employed two safeguards: (1) restricting task creation to **PhD holders and senior practitioners with 5+ years of domain experience**, and (2) conducting **rigorous manual screening** by our team to verify quality and complexity.
>
> More importantly, in our evaluation framework, **task difficulty is not determined by the task itself alone**. The RACE framework as a whole ensures appropriate difficulty through its design: both the **adaptive criteria** and **reference reports** dynamically adjust the evaluation standards for each task, thereby influencing the actual difficulty experienced by DRAs. This holistic approach ensures that each task poses a meaningful challenge while remaining evaluable through our automated framework.
>
> ### ●  **To W5 & Q3: RACE Criteria Generation**
>
> We clarify that this is described in Section 3.1.1. The RACE criteria are **dynamically generated per task** (i.e., each of the 100 tasks has its own tailored criteria and weights). However, **once generated for a task, these criteria remain fixed** across all DRA evaluations for that task. This design ensures:
>  - Fair comparison: all DRAs are evaluated against identical, task-specific criteria;
>  - Scalability: future work can easily extend the benchmark by generating criteria for new tasks without re-evaluating existing ones.
> To clarify this important distinction, we have added an explicit statement in the **revised paper**.

---

> > ### Author Response · Authors · 2025-11-22
> >
> > ### ●  **To W6: Citation Coverage**
> >
> > Citation coverage is a valuable dimension, but contains **subjective components that are difficult to automate**. In our paper, FACT prioritizes **objectivity and automation** by anchoring on DRA-provided citations: it extracts statements paired with their citations, fetches the cited URL content, and determines citation accuracy—a relatively objective and fully automated process. In contrast, the referenced paper (arXiv:2411.17375) defines coverage as (# sentences where all claims are cite-supported) / (# total sentences). For DRA-generated reports, the denominator is difficult to determine: which sentences require citations? Some statements may derive from external knowledge or logical reasoning without citations, yet remain valid. This ambiguity makes the metric **less actionable** across diverse DRAs. Moreover, the paper's evaluation requires **human annotators to manually identify claims and verify citation support** for each sentence—difficult to automate at benchmark scale. FACT's focus on citation precision and effective citation count provides objective, automated, and actionable evaluation essential for reproducible benchmarks.
> >
> > ### ●  **To W7 & Q5: Gemini as Reference Reports**
> >
> > **Why Gemini as reference:** Our RACE framework requires a high-quality reference report for each task. However, inviting domain research teams to produce expert-level reports for each task is impractical and unscalable. We must select the highest-quality and most transparent DRA from available systems. We chose **Gemini-2.5-Pro Deep Research** because it offers **greater community transparency** compared to alternatives like OpenAI and Grok (free-tier accessible with reasonable limits, whereas OpenAI's free quota is too restrictive). This is documented in Section 4.1.
> >
> > As the reviewer notes, this inevitably raises concerns about model bias. To address this, we conducted **comprehensive human consistency experiments** to validate RACE's robustness. As shown in Section 4.3.4 and Appendix "ANALYSIS OF RACE ROBUSTNESS", we demonstrate that our framework **minimizes model bias impact**: changing the reference or judge does not affect the final conclusion (rankings remain stable). Gemini as judge achieves the **highest human consistency** at an acceptable cost. These results confirm that RACE mitigates bias concerns effectively.
> >
> > ### ●  **To Q4: Judge LLM Calibration**
> >
> > All judge models in our experiments use **identical prompts** for both criteria generation and final evaluation, ensuring fair comparison. If the reviewer's question refers to whether we optimized prompts for each judge or setting: we did not, as prompt engineering is not the core of our work. Our focus is on proposing an **innovative and robust evaluation framework** for end-to-end DRA assessment. The consistency of prompts across all judges strengthens the validity of our robustness analysis.

---

### Official Review · Reviewer_Yinh · 2025-11-01

**Soundness:** 3
**Presentation:** 4
**Contribution:** 3
**Rating:** 8
**Confidence:** 5

**Summary:**

The paper introduces **DeepResearch Bench**, a new benchmark consisting of 100 carefully curated, PhD‑level research tasks spanning 22 domains. It proposes two automated evaluation frameworks:

* **RACE** – a reference‑based, dynamically weighted “LLM‑as‑judge” protocol that generates task‑specific criteria and dimension weights (Comprehensiveness, Insight, Instruction‑following, Readability) and scores reports relative to a high‑quality reference.
* **FACT** – an automated pipeline that extracts statement‑URL pairs from agent reports, deduplicates them, and judges factual support using a separate judge LLM, yielding Citation Accuracy and Average Effective Citations per task.

The authors validate RACE and FACT by (i) extensive human consistency experiments (pairwise agreement, Pearson/Spearman correlations), (ii) robustness analyses (reference selection, length inflation, judge model choice), and (iii) comparisons across a wide range of commercial Deep Research Agents (DRAs) and LLMs with search tools. All data, prompts, and code are released. The paper makes a solid engineering contribution by releasing a well‑designed benchmark and novel, human‑aligned evaluation frameworks for Deep Research Agents. The methodology is sound, the experiments are extensive, and the analyses address many potential pitfalls. However, the reliance on proprietary judges, limited benchmark scale, and lack of statistical significance reporting temper the impact. Overall, the work is a valuable resource for the community and merits inclusion in the conference, albeit not as a flagship oral contribution.

**Strengths:**

1. **Clear Gap Identification** – The paper convincingly argues that existing benchmarks either target isolated capabilities (browsing, code) or single‑domain research, leaving a need for a comprehensive DRA evaluation suite.
2. **Benchmark Construction Pipeline** – Leveraging a large in‑house query log (≈96 k queries) and a taxonomy‑driven filtering/classification pipeline yields a realistic distribution of tasks. Expert‑authored tasks and a bilingual (50 EN/50 ZH) set improve ecological validity.
3. **Innovative Evaluation Design** – RACE’s dynamic weighting and reference‑based scoring address known pitfalls of static rubrics and absolute LLM scoring (e.g., length bias, generic high scores). FACT provides a concrete, interpretable measure of citation grounding, which is highly relevant for research‑assistant agents.
4. **Human‑Alignment Validation** – The human consistency study is thorough: 70+ annotators, pairwise agreement (PAR = 71 % for RACE vs. 59 % for a vanilla prompt), and high Pearson correlations (OPC ≈ 0.99). The filtered‑task analysis further strengthens confidence.
5. **Robustness Analyses** – Sensitivity to reference choice, report length inflation, and judge LLM (closed‑source vs. open‑source) is explored, showing that rankings are stable.
6. **Reproducibility Commitment** – All prompts, hyper‑parameters, and scripts are released; the paper details cost per evaluation and provides open‑source judge alternatives.
7. **Ethical Considerations** – The authors appropriately describe anonymization of query logs, fair compensation of annotators, and discuss potential misuse of generated reports.

**Weaknesses:**

1. **Scale and Diversity Limitations** – 100 tasks, while high quality, may still be insufficient to capture the full variance of real‑world research queries, especially for emerging sub‑domains (e.g., AI‑ethics, climate modeling). This limits statistical power for fine‑grained comparative analysis.
2. **Reliance on Proprietary Judge LLMs** – The primary RACE and FACT evaluations depend on Gemini‑2.5‑Pro/Flash APIs. Although the authors provide open‑source alternatives (Qwen3) and show similar rankings, the absolute scores and some correlation metrics dip noticeably, raising concerns about reproducibility for researchers without access to the same commercial models.
3. **Citation Extraction Fragility** – FACT assumes that agents emit citations in a parsable format; the paper notes that several DRAs (Claude Research, Kimi) could not be evaluated because of UI‑specific citation styles. This suggests that FACT may penalize agents for UI design rather than factual grounding.
4. **Baseline Coverage** – The experimental section focuses on commercial DRAs and a few open‑source agents (LangChain ODR). There is no systematic evaluation of baseline retrieval‑only pipelines (e.g., traditional IR + summarizer) that could contextualize the performance gap.
5. **Statistical Reporting** – While human‑consistency numbers are presented, confidence intervals or statistical significance tests for the differences between agents are absent. It is unclear whether the observed gaps (e.g., Gemini‑2.5‑Pro vs. OpenAI DRA) are robust.
6. **Potential Conflict of Interest** – Many evaluated agents are from the same ecosystem (Google, OpenAI, Perplexity) that also provide the judge LLMs. The paper does not discuss mitigation of possible bias introduced by using a judge model that may be tuned toward its own family’s output style.
7. **Limited Discussion of Failure Cases** – The analysis of low citation accuracy for high‑citation agents (e.g., Gemini‑2.5‑Pro) is brief; deeper error analysis (hallucinated citations, URL unreachable, mis‑matched statements) would improve understanding of trade‑offs.

**Questions:**

1. **Task Sampling Variability** – How sensitive are RACE scores to the specific set of 100 tasks? Have you performed a leave‑one‑out or bootstrapped analysis to estimate variance across different task subsets?
2. **Open‑Source Judge Calibration** – When using Qwen3 as the judge, did you fine‑tune prompts or temperature settings to match the proprietary judge’s behavior? Could the performance gap be reduced with prompt engineering?
3. **Citation Format Normalization** – Could you elaborate on the preprocessing pipeline that normalizes citations across agents? Is there a risk that aggressive cleaning removes legitimate citation cues, thereby affecting FACT scores?
4. **Statistical Significance** – Did you compute confidence intervals for the pairwise agent comparisons (e.g., via bootstrap of human scores)? If not, could you add these to substantiate claims of superiority?
5. **Ethical Review of Web Scraping** – FACT retrieves full web pages via Jina Reader. Were any of the source pages subject to robots.txt restrictions or copyright considerations, and how were these handled?

---

> ### Author Response · Authors · 2025-11-22
>
> We thank the reviewer for the encouraging feedback and helpful suggestions.
>
> # Weakness
>
> ### ●  **To W1**
>
> **On Dataset Scale:** We acknowledge this trade-off. To ensure task quality (expert-crafted, grounded in real demand), we limited dataset size. Leading benchmarks (xbench-DeepSearch, Mind2Web 2) adopt similar scales.
>
> **On Sub-domain Granularity:** Our goal is **end-to-end evaluation of overall DRA capabilities** across diverse research scenarios, rather than fine-grained within-domain analysis. Instead, we hope this work inspires domain-specific benchmarks for emerging sub-fields where deeper focus is needed.
>
> ### ●  **To W2: Reliance on Proprietary LLMs**
>
> We address this in Section 4.3.3 and Appendix. We provide **Qwen3-235B-Thinking (Apache 2.0)** as an open-source alternative. While its human-consistency metrics are slightly lower than proprietary judges (Table 4), **the relative ranking of DRAs remains identical**—which is what matters for reproducible comparative evaluation. Researchers can use Qwen3 to evaluate their own DRAs and obtain reliable, consistent rankings.
>
> ### ●  **To W3: Citation Extraction Fragility**
>
> We want to clarify that this reflects a **product design limitation, not a FACT framework flaw**. Claude Research and Kimi do display citations in their UI, indicating they possess citation capabilities. However, they do not provide exportable, machine-parsable citation formats. The "--" marks in Table 1 indicate **missing data due to inaccessibility**, not a penalty for their citation quality. We note this limitation in Section 4.2.2 and encourage vendors to adopt standardized, exportable citation formats to enable comprehensive evaluation.
>
> ### ●  **To W4: Baseline Coverage**
>
> We respectfully clarify that we **did include baseline comparisons**: the "LLM with Search Tools" represents traditional retrieval-augmented approaches. As noted in Section 4.3.1, these models "struggle to compete with modern DRAs," providing clear performance contextualization.
>
> Traditional IR+summarizer pipelines **focus on offline single-round retrieval, which is not suitable as a baseline for DRA evaluation.**
>
> Considering the reviewer's concern, we have conducted additional evaluations and included results in the **revised paper**: (1) **DeepResearcher**, an early DRA system that resembles the traditional IR+summarizer pipeline but with support for wild web search rather than offline retrieval; (2) **Tongyi DeepResearch**, the most recognized fully open-source DRA to date. Our analysis reveals that DeepResearcher (10.77) struggles with long-form report generation, highlighting the gap between early retrieval-based approaches and modern DRAs, while Tongyi DeepResearch (40.46) achieves comparable performance to some proprietary agents. We believe these are the most appropriate baselines for the deep research paradigm. If the reviewer has more specific suggestions, we welcome further discussion.
>
> ### ●  **To W5: Statistical Reporting & Q4: Statistical Significance**
>
> We thank the reviewer for this valuable suggestion. We have added paired t-tests with 95% confidence intervals in the **revised paper** (new Appendix section "Statistical Analysis of Performance Differences") to substantiate the observed performance differences. The statistical analyses confirm that Gemini-2.5-Pro DR and OpenAI DR significantly outperform the other agents, while Claude/Kimi/Doubao show no significant pairwise differences.
>
> ### ●  **To W6: Potential Conflict of Interest**
>
> Our RACE framework is designed to mitigate judge-related biases through its reference-based, adaptive criteria mechanism. Given that Judge LLMs must inevitably come from one of these frontier model families, we validated our design choice through comprehensive human consistency experiments (Section 4.3.3).As demonstrated in Appendix "Analysis of RACE Robustness," the evaluation results are robust to Judge model selection—**DRA relative rankings remain identical** across different Judge LLMs (e.g., Gemini-based vs. Qwen3-based). This indicates that same-family bias does not significantly impact the evaluation outcomes under our framework.
>
> ### ●  **To W7: Limited Discussion of Failure Cases**
>
> This is a valuable suggestion. We have conducted preliminary analysis on citation failure patterns. For instance, we found that LangChain ODR frequently cites sources unrelated to its statements. By tracing its execution logs, we identified that this stems from its multi-agent architecture with SLM-based summarization, which causes citation mapping misalignment during information aggregation. Given that comprehensive failure mode analysis across different DRAs requires substantial human resources=, we will conduct systematic analysis and include these findings in a future version of the paper.

---

> > ### Author Response · Authors · 2025-11-22
> >
> > # Questions
> >
> > ### ●  **To Q1: Task Sampling Variability**
> >
> > We conducted leave-one-out (LOO) analysis in the **revised paper** (same Appendix section as W5). All agents maintained their ranks in 100% of LOO iterations, confirming that rankings are highly stable and not driven by any single task.
> >
> > ### ●  **To Q2: Open-Source Judge Calibration**
> >
> > To address the reviewer's question, we conducted additional experiments to Qwen-as-Judge. Our default settings are temperature=1.0 and thinking-budget=16k. See the table below.
> >  - When fixing temp and varying budget, we found that 16k significantly outperforms 8k, while 32k shows no significant improvement despite consuming more computational resources.
> >  - When fixing budget and adjusting temperature, we found that the default 1.0 performs best.
> >
> > **Ablation on Thinking Budget (Temperature=1.0 fixed):**
> >
> > | Budget | PAR | OPC | FAP | FAS | Overall |
> > |--------|-----|-----|-----|-----|---------|
> > | 8k | 69.33 | 84.01 | 56.42 | 50.28 | 65.01 |
> > | 16k (default) | 70.78 | **84.47** | 56.80 | 56.94 | 67.25 |
> > | 32k | **71.44** | 81.44 | **58.51** | **58.17** | **67.39** |
> >
> > **Ablation on Temperature (Budget=16k fixed):**
> >
> > | Temperature | PAR | OPC | FAP | FAS | Overall |
> > |-------------|-----|-----|-----|-----|---------|
> > | 0.0 | 69.22 | 80.26 | 55.77 | 51.98 | 64.31 |
> > | 1.0 (default) | **70.78** | **84.47** | 56.80 | **56.94** | **67.25** |
> > | 1.9 | 70.18 | 82.27 | **58.00** | 54.80 | 66.31 |
> >
> > And we found that prompt engineering has minimal impact on consistency metrics. In conclusion, **adjusting prompts and inference parameters does not substantially affect results**—there remains an inherent gap between open-source and proprietary models that cannot be fully bridged.
> >
> > ### ●  **To Q3: Citation Format Normalization**
> >
> > We clarify that **citation cleaning only applies to RACE evaluation, not FACT**. As noted in Section 4.1, RACE cleans citation formatting to avoid format-induced bias and prevent reports from exceeding context limits. **FACT evaluation uses complete, original citations**, no cleaning is performed, ensuring all legitimate citation cues are preserved for accurate verification.
> >
> > ### ●  **To Q4**
> >
> > See above.
> >
> > ### ●  **To Q5: Ethical Review of Web Scraping**
> >
> > We appreciate the reviewer’s valuable suggestion. We fetch only publicly accessible pages and our fetcher (Jina Reader) now supports an HTTP header x-robots-txt to enforce robots.txt policies (i.e., skip pages disallowed by robots). We will enable this option and document it in our publicly released code so that robots compliance is explicitly enforced in our pipeline.

---

### Author Response · Authors · 2025-11-22
**Official Comment to All Reviewers**

We sincerely thank all reviewers for the valuable feedback and constructive suggestions. We have carefully addressed each concern and prepared a **revised paper** with all modifications **highlighted in blue** for easy identification.

---

### Meta-Review · Area_Chair_yx3p · 2026-01-08

**Summary:**

The paper introduces DeepResearch Bench, a benchmark designed to evaluate Deep Research Agents (DRAs) on open-ended, complex research tasks. It consists of 100 expert-crafted tasks spanning 22 domains (50 English, 50 Chinese). The authors propose two automated evaluation frameworks: RACE (quality assessment using criteria generation and reference reports) and FACT (citation accuracy and verification). The submission includes experiments on various commercial and open-source agents, validated by human consistency studies.

**Reviewer Concerns:**

**Addressed:**

**Baselines:** Reviewer xj7Q and Yinh requested open-source baselines to better contextualize performance. The authors added evaluations for DeepResearcher and Tongyi DeepResearch in the revision.

**Judge Bias & Open Source Alternatives:** Reviewers Yinh and oeAc expressed concern about using Gemini as both the reference generator and the judge. The authors conducted robustness analyses showing that rankings remain stable across different judges and added experiments using an open-source judge (Qwen3).

**Statistical Significance:** Reviewer Yinh requested statistical tests. The authors added pairwise t-tests with 95% confidence intervals to the revised paper.

**Clarity & Formatting:** Reviewers noted fragmented method descriptions and formatting issues. The authors moved FACT calculation details to the main text and fixed prompt formatting.



**Outstanding:**

**Dataset Scale:** Multiple reviewers (Yinh, oeAc, 8pm9) noted that 100 tasks is a small sample size for a comprehensive benchmark, limiting fine-grained domain analysis. The authors argue this is a trade-off for expert curation, but the limitation regarding statistical power for sub-domains remains.


**Human Evaluation Scope:** Reviewer oeAc noted that human consistency validation was performed only on the Chinese subset of tasks. While the authors clarified that the *main* evaluation results use the full set , the validation of the metric itself relies on the Chinese subset due to resource constraints, which is a valid limitation for the English portion of the benchmark.

**Reviewer Scores:**

**Reviewer Yinh (Current: 8):** will 8. The reviewer was already positive, and the authors addressed their requests for statistical testing and open-source judge ablation.


**Reviewer xj7Q (Current: 6):** will 6. The authors addressed the primary weakness regarding missing baselines and clarified task selection/difficulty. It makes the submission stronger but shouldnt change the score to another level.


**Reviewer oeAc (Current: 4):** will 6. The main soundness concern was a misunderstanding that the main results might be based only on the Chinese split. The authors clarified that main results cover all 100 tasks, which should resolve the reviewer's primary hesitation.

**Reviewer 8pm9 (Current: 8):** will 8. This reviewer found no major weaknesses.

---

### Decision · Program_Chairs · 2026-01-26

Accept (Poster)